# *TEFM* variants impair mitochondrial transcription causing childhood-onset neurological disease

Mutations in the mitochondrial or nuclear genomes are associated with a diverse group of human disorders characterized by impaired mitochondrial respiration. Within this group, an increasing number of mutations have been identified in nuclear genes involved in mitochondrial RNA biology. The *TEFM* gene encodes the mitochondrial transcription elongation factor responsible for enhancing the processivity of mitochondrial RNA polymerase, POLRMT. We report for the first time that *TEFM* variants are associated with mitochondrial respiratory chain deficiency and a wide range of clinical presentations including mitochondrial myopathy with a treatable neuromuscular transmission defect. Mechanistically, we show muscle and primary fibroblasts from the affected individuals have reduced levels of promoter distal mitochondrial RNA transcripts. Finally, *tefm* knockdown in zebrafish embryos resulted in neuromuscular junction abnormalities and abnormal mitochondrial function, strengthening the genotype-phenotype correlation. Our study highlights that *TEFM* regulates mitochondrial transcription elongation and its defect results in variable, tissue-specific neurological and neuromuscular symptoms.

Mitochondria play an essential role in regulating cellular homoeostasis as they are the main source of ATP produced by oxidative phosphorylation (OXPHOS) and they host a number of biosynthetic pathways. Thirteen essential subunits of the OXPHOS system are synthesized within the organelle. Mitochondria maintain and express the mitochondrial DNA (mtDNA) that codes for these polypeptides, as well as the full set of mitochondrial (mt-) tRNAs and two mt-rRNAs. All remaining mitochondrial protein components are encoded by the nuclear genome and imported into the mitochondria from the cytoplasm. This includes components of the mitochondrial transcription machinery such as the mitochondrial RNA polymerase (POLRMT), transcription initiation factors (mitochondrial transcription factor A—TFAM and mitochondrial transcription factor B2—TFB2M) and transcription elongation factor (TEFM). More than 50 nuclear-encoded mitochondrial proteins involved in mitochondrial gene expression have been linked to heritable disorders[1–3].

Transcription of human mtDNA initiates in the main non-coding region (NCR) from the L-strand (LSP) and H-strand (HSP) promoters and produces two, nearly genomic-length polycistronic precursor RNAs. LSP controls the transcription of eight mt-tRNAs and the *ND6* gene, whereas HSP generates a transcript that encodes the remaining twelve protein-coding genes, two mt-rRNA and fourteen mt-tRNA[4]. Mitochondrial transcription in human mitochondria is driven by DNA-dependent-RNA polymerase POLRMT, which is structurally similar to RNA polymerases in T3 and T7 bacteriophages[5,6]. However, in contrast to bacteriophage polymerases, which can recognize promoters without auxiliary proteins, POLRMT requires additional factors. The association of POLRMT to TFAM and TFB2M is required for transcription initiation. TFAM is a DNA-binding protein that recruits POLRMT to the promoter for transcription activation, but also packages DNA in the nucleoid, whereas the key function of TFB2M is to modify the structure of POLRMT to induce promoter melting[7–11].

✉e-mail: michal.minczuk@mrc-mbu.cam.ac.uk; rh732@medschl.cam.ac.uk

TEFM is required by POLRMT for the elongation stage[12]. In vitro, recombinant TEFM strongly promotes POLRMT processivity and stimulates the formation of longer transcripts[13]. Depletion of TEFM in cultured cells or in vivo in a conditional *Tefm*-knockout mouse model leads to a drastic reduction in promoter-distal transcription elongation products driven from both HSP and LSP[12,14]. Recent data also indicate the involvement of TEFM in mtRNA maturation, as analysis of the transcriptome from *Tefm* knockout mice revealed accumulation of unprocessed transcripts[14]. Mitochondrial transcription and replication are functionally linked. LSP-derived transcripts are often prematurely terminated around the conserved sequence block II (CSB II) of the mtDNA major non-coding region (NCR) via the formation of a G-quadruplex structure between nascent RNA and the non-template strand of mtDNA. It has been proposed that these prematurely terminated RNA molecules play a key role in priming DNA replication, as multiple RNA to DNA transition sites cluster around CSB II[15,16]. Stimulation of POLRMT processivity by TEFM in vitro was shown to prevent the formation of G-quadruplexes that abolishes the progression of the elongation complex at CSB II[13,17]. Therefore, it has been suggested that the capability of TEFM to inhibit premature transcription termination functions as the switch from replication to the transcription of the LSP-derived primary transcript[17]. However, in vivo data argue against this model, as inactivation of TEFM does not lead to upregulation of mtDNA replication initiation and the short LSP-derived transcripts cannot be used for mtDNA replication[14]. Recent structural work showed that TEFM interacts with POLRMT via its C-terminal domain. TEFM contains a pseudonuclease core that forms a 'sliding clamp' around the mtDNA downstream of the transcribing POLRMT[8].

While a large number of mitochondrial disorders have been reported as a consequence of defects in components of the mtDNA replication machinery[18], pathogenic variants in the mitochondrial transcription components have only just started to emerge. Very recently variants in *POLRMT* have been associated with mitochondrial dysfunction and a broad spectrum of neurological presentations[19]. A homozygous missense variant in the *TFAM* gene was identified in one family that led to neonatal onset of rapidly progressive liver failure, resulting in death in infancy. However, the follow up studies identified decreased mtDNA copy number and perturbations in the number and size of mitochondrial nucleoids in samples from the affected individuals, suggesting that the mutation affected mtDNA maintenance, consistent with TFAM's function in packaging mtDNA, in addition to transcription activation[20].

In the present work, we report the identification of seven *TEFM* variants (four missense, two frameshift and one in-frame 2-amino acid deletion) in seven individuals from five unrelated families who present with mitochondrial respiratory chain deficiency and a wide range of infantile or childhood-onset neurological and neuromuscular symptoms, due to abnormal mitochondrial transcription.

## Results

### Summary of clinical features of the investigated patient cohort
The clinical presentation and investigations of the 7 patients are listed in Supplementary Table 1 and pedigrees of investigated families in Fig. 1. Detailed case reports are provided in the Supplementary Material. The onset of symptoms ranged from a few hours after birth to 5 years of age. All parents were healthy and family history was compatible with an autosomal recessive inheritance pattern, with two affected siblings in two families, consanguinity was only reported in one family (Fig. 1). The phenotype was strikingly variable; some patients had multisystem presentations ranging from a rapid neonatal metabolic decompensation with severe lactic acidosis, hypoglycaemia, neurological impairment, liver failure and death at 1 month of age (P3), to severe, progressive epileptic encephalopathy with myoclonic jerks, psychomotor regression, vomiting, ptosis and fluctuating ophthalmoparesis leading to death at 17 months of age (P4). The less severe

patients were ambulatory with slowly progressive developmental delay, autism spectrum disorder and mild intellectual disability in a 5-year old girl with onset at 2.5 years of age (P5), non-specific mild developmental delay and ataxia in two siblings (P6, P7), one of whom also presented with optic atrophy and fluctuating proximal muscle weakness, ophthalmoparesis and epilepsy (P1, P2).

Brain imaging (MRI) was normal in P1 and P5, but showed some mild non-specific changes in the other patients such as microcephaly, white matter lesions (P2), lactate peak (P3), loss of white matter and cortical atrophy (P4), cerebellar atrophy, demyelination and thin corpus callosum (P6, P7) (Fig. 2a). While P1, P2 and P4 had seizures of variable severity, P3 died at 1 month of age without clinical signs of epileptic activity, and P5, P6 and P7 had no clinical or electrophysiological evidence of epilepsy.

Two brothers (P1, P2) had a childhood-onset fluctuating muscle weakness and fatigability, and were clinically diagnosed as possible congenital myasthenic syndrome. The muscle weakness was more progressive in P1 who had myopathic pattern on EMG with fatty replacement of proximal and distal leg muscles (Fig. 2b) and an abnormal repetitive stimulation (Fig. 2c), suggesting a defect of neuromuscular transmission. While both siblings had seizures, P2 had additional features of moderate intellectual disability with tremor and ataxia. P1 showed significant improvement in muscle strength and ambulation with Salbutamol (6 mg/day) for several months.

Skeletal muscle biopsy showed mitochondrial abnormalities with SDH hyper-reactivity and COX negative fibres (Fig. 2d) and a combined defect in the activities of OXPHOS enzyme complexes I and IV in P1, P2 and P4. The muscle biopsy of P3 taken during infancy was not interpretable and the other patients did not have a muscle biopsy.

While most patients had some degree of muscle, brain and liver involvement, the clinical variability was very broad in our cohort. This may in part be explained by the severity of the different variants, however, there was a significant variability even within families (P1/P2 and P6/P7 had different clinical presentations).

### Identification of TEFM variants
To identify the genetic cause for the observed phenotypes, we performed whole exome sequencing (WES) or whole genome sequencing (WGS) in the patients and their parents and identified biallelic novel missense ($n = 4$), frameshift ($n = 1$), splice site ($n = 1$) or in-frame deletion ($n = 1$) variants in the *TEFM* gene (ENST00000581216.6_1/ NM_024683.4; Supplementary Table 1, Fig. 1, Fig. 3a). All missense variants detected showed strong evolutionarily conservation, except for p.(Lys188Arg), which showed little conservation (188Arg in the mouse), but may affect splicing as predicted by the Human Splicing Finder (HSf, http://umd.be/Redirect.html) and the spliceAI (https://spliceailookup.broadinstitute.org/) programmes (Fig. 3b). However, consent was not obtained for a skin biopsy which may have allowed for confirmation of abnormal splicing in primary fibroblasts. Most of these variants were extremely rare or not reported in gnomAD (https://gnomad.broadinstitute.org/) and predicted a damaging effect using Polyphen-2[21]. The compound missense variants found in P5 (Family 4) p.(Arg34Trp) (1.64e−4; 41 heterozygotes in 249,466) and p.(Lys188Arg) (4.27e−4; 120 heterozygotes in 280,950) have relatively high allele frequency in gnomAD, but no homozygotes were reported. The analysis of mRNA splicing in P4 (Family 3), carrying an intronic variant c.32-14A>G (paternal) revealed skipping of exon 2 affecting approximately half of *TEFM* transcripts and predicting a frameshift and premature stop codon (p.Glu11Glyfs*2). Of the remaining transcripts that do encode exon 2, the maternally inherited c.484_489delGAAAGA variant was present in 70% of transcripts, consistent with allelic imbalance. The parents were healthy in all families and carried one heterozygous variant in each case. The second heterozygous variant detected in P4, c.484_489delGAAAGA, which results in the deletion of two amino acids (p.(Glu162_Arg163del)), was also detected in

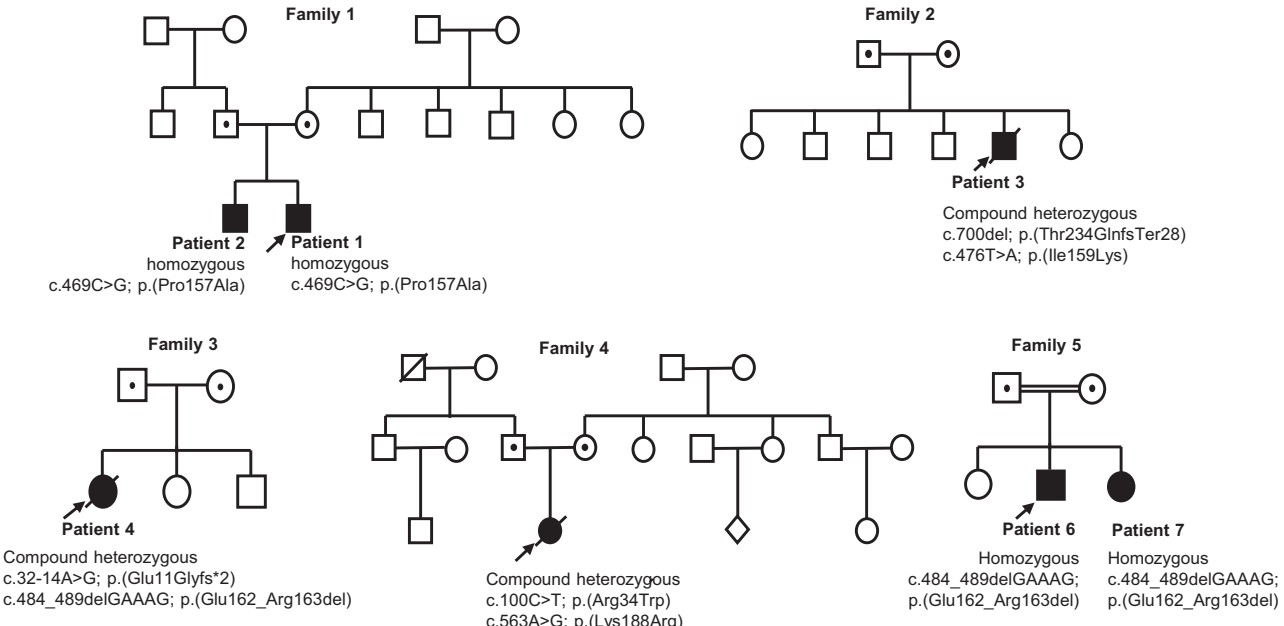

**Fig. 1 | Pedigrees.** Pedigrees of the five families with associated *TEFM* variants.

homozygosity in Family 5 (P6 and P7). Furthermore, we demonstrate that the 288c.484_489delGAAAGA variant leads to reduced activity of mitochondrial complex I in fibroblasts of P4 (Supplementary Fig. 1).

## In silico characterization of TEFM variants

Next, we set out to model the effect of the detected variants on TEFM function. We used the structure of the TEFM dimer associated with POLRMT[22]. However, in this the structural model of TEFM protein is not complete and lacks the N-terminal 145 amino acids, including the MTS and hairpin-hairpin-helix domain (Fig. 3c, d). All detected missense mutations that could be modelled on the existing structures are predicted to be localized in a solvent-exposed surface, opposite to the surface that interacts with POLRMT. The Pro157 residue is located near the dimerization interface of TEFM and, despite the lack of an evident hydrophobic environment in the surrounding region, its change to alanine may impact on the dimerization of this protein. This is supported by the participation of Ser210 from one of the TEFM monomers in the local environment of Pro157 of the other monomer. Furthermore, Pro157 is located at the end of a helix and may contribute to the delimitation of its secondary structure due to the conformational rigidity of proline conferred by its cyclic structure, which is not compatible with the angular requirements for helices. Hence, the p.(Pro157Ala) TEFM protein may present conformational differences compared to wild-type. The structural impact of the p.(Pro157Ala) variant is relatively limited, as alanine has a very short side chain, which is not expected to create a steric clash. Also, the nature of the side chains of proline and alanine are relatively similar (short and aliphatic), so there are no dielectric changes in this region of the protein. Taken together, there is no clear evidence to support the pathogenicity of the p.(Pro157Ala) TEFM variant from this structural analysis (Fig. 3d). The Ile159 residue may also have a role in the dimerization of TFEM, or the stabilization of the interface. Substitution of isoleucine to a lysine is expected to disrupt the environment by adding an electric charge and introducing a bulky side chain. Furthermore, in the vicinity of Ile159, the side chain of Arg163 forms a polar interaction with the backbone of His271, which is potentially relevant for the overall structure of TEFM. Given this, p.(Ile159Lys) would introduce a positive charge and has a longer side chain, which could disrupt the Arg163-His271 interaction (Fig. 3d). The loss of Glu162 and Arg163 in addition to impacting on the global

structure as mentioned above, is also predicted to perturb a helical element which, although solvent-exposed, could have a role in maintaining long-range sequence interactions (Fig. 3d). The side chain of Lys188 is exposed in the analyzed structure and is located in a loop between beta-strands, next to Arg187. Since the detected variant alters lysine to arginine, i.e. an amino acid with a similar charge and size, any predictions as to damaging properties of this change are difficult. On the other hand, the change to two consecutive arginines could result in stacking interactions, affecting the local environment. Taken together, based on in silico analysis, the prediction of pathogenicity is challenging and, therefore, we set out to investigate the effects of the detected variants in the following in vitro and cellular studies.

## Analysis of TEFM levels and OXPHOS function in affected individuals

In order to get further insights into the pathogenicity of the detected *TEFM* variants, we assessed the TEFM protein stability in patient-derived primary skin fibroblasts. The steady-state level of TEFM was substantially reduced in P2 and P3 fibroblasts as compared to the controls, consistent with our prediction of p.(Pro157Ala) (P2) and p.(Ile159Lys) (P3) impacting protein conformation and stability (Fig. 4a). However, further analysis of primary dermal fibroblasts from P2 and P3 by western blotting showed no substantial changes in steady-state levels of the OXPHOS components (Fig. 4b, c). Notwithstanding, a similar analysis of the OXPHOS subunits in fibroblasts of P4 showed reduced levels of NDUFB8 (complex I) and normal levels of the other OXPHOS components (Supplementary Fig. 2a). Quantitative proteomics was used to confirm the levels of TEFM protein in P4 and assess the impact on complex I (Fig. 4d, e, Supplementary Dataset). As can be seen in Fig. 4d, P4 fibroblasts show decreased amounts of TEFM protein (−2.2 fold-change) relative to controls. It should be noted that TEFM protein was quantified from a single peptide sequenced by MS/MS in all samples (Supplementary Fig. 2d). Relative complex abundance profiling revealed a 25% reduction in the abundance of complex I subunits in P4 relative to controls. Complex II and complex IV presented a 12% and 7% reduction, respectively, while the mitoribosome presented a 10% increase in relative abundance (Fig. 4e). Furthermore, the analysis of OXPHOS activities (complexes I and IV) in fibroblasts from P4 showed a 35% reduction in complex I, but no

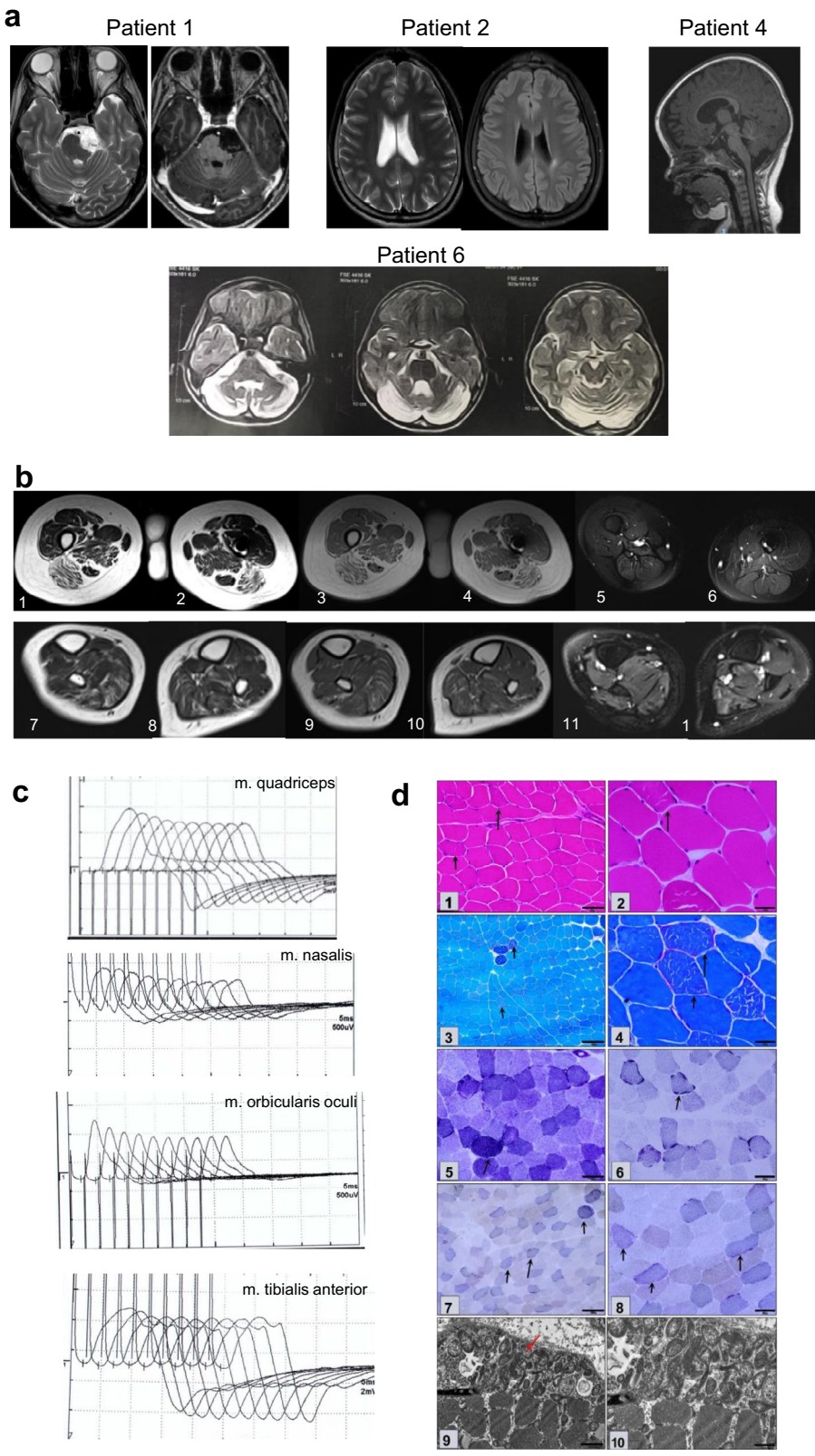

difference in complex IV (Supplementary Fig. 2b, c). The lack of OXPHOS phenotype observed in the patient fibroblasts, except P4, prompted us to analyze patient tissues affected by the disease. To this end we investigated the available quadriceps muscle (*vastus lateralis*) samples from the patients. In this analysis we detected markedly reduced protein levels of subunits from RC complexes I, III and IV in muscle of P1 & P2 compared to control samples (Fig. 4f). These data

show that p.(Pro157Ala) (Family 1, P1 and P2) and p.(Ile159Lys) (Family 2, P3) variants perturb the stability of the TEFM protein and in the case of P1 and P2 lead to a reduced stability of the OXPHOS components in the skeletal muscle. Furthermore, these results also demonstrate that the c.484_489delGAAAGA variant identified in Family 3 (P4), which is also found in Family 4 (P6 and P7), leads to reduced activity of mitochondrial complex I.

**Fig. 2 | Brain and muscle imaging, muscle histology and repetitive stimulation.**
**a** Patient 1: an epidermoid cyst in pre-pontine cistern. Axial T2 W image and post contrast enhancement of the lesion. Patient 2: Axial T2W and FLAIR images show focal hyperintense area in left periventricular region. Patient 4: loss of white matter and cortical atrophy on FLAIR images. **b** Muscle MRI images of patient 1. Images in the first row are at mid-thigh level. T2 W axial images (1, 2), T1 W images (3, 4) show atrophy with fatty replacement of bilateral biceps femoris, semimembranosus & semitendinosus. STIR images (5, 6) showing no signal change in the involved muscles. Images in the second row are at calf level. T2 W axial images (7, 8), T1 W axial images (9, 10) are showing fatty replacement of the gastrocnemius & soleus. Axial STIR images (11, 12) show no significant signal changes. **c** Repetitive standard 3 Hz slow nerve stimulation (5th wave decrement) in Patient 1. Quadriceps

demonstrated 21.1% decrement, nasalis shows 30.3%, orbicularis oculi shows 26.5% and no significant decrement detected from the tibialis anterior muscle.
**d** Quadriceps muscle biopsy of patient 1: HE (1–2) showed polygonal fibres with peripherally placed nuclei and minimal variation in fibre size. No evidence of vacuoles, some occasional fibres showed internalized nuclei. Masson's Trichrome (3) showed no endomysial fibrosis. Modified Gomori (4) trichrome stain showed thin subsarcolemmal accumulation while NADH stain (5–6) showed increase in subsarcolemmal accumulation. COX-SDH stain (7–8) detected COX deficient fibres in more than 50% of fibres. Electron microscopy (9–10) showed abnormal mitochondrial structure and cristae. Scale bar: 1: 50 μm; 2,4: 20 μm; 3,5,6: 100 μm; 7-8: 1 μm.

## Analysis of mitochondrial DNA copy number in affected individuals

To investigate the effect of *TEFM* variants on mitochondrial replication we investigated the mtDNA copy number in patient fibroblasts. We detected a slight elevation in mtDNA levels in samples from P2 and P3 (Fig. 4g), as compared to control fibroblasts, while no differences were detected in P6 (Supplementary Fig. 2e). Analysis of mtDNA copy number in quadriceps muscle (*vastus lateralis*) of P1 & P2 detected a substantial elevation in mtDNA copy number compared to control muscle (Fig. 4h). Given that the mitochondrial mass is not increased in patient cells (Fig. 4b, f), the observed increase in mtDNA content may not result from the compensatory response due to perturbed mitochondrial gene expression, as observed previously[23]. In this view, these results indicate that mutations in *TEFM* may result in more frequent termination at CSBII.

## Analysis of mitochondrial transcriptome in affected individuals

Next, we set out to investigate the transcriptome in TEFM patient samples. To this end, we analyzed mRNA levels in the available patient skeletal muscle samples (P1 and P2) and fibroblasts (P2 and P3) and healthy controls by genome-wide RNA-Seq. Pathway analysis of the differentially expressed genes in skeletal muscle of P1 and P2 showed that mitochondria-related pathways were specifically affected in the muscle samples (Supplementary Fig. 3a, b). Further analysis showed a reduction of mRNAs for all mitochondrial encoded genes, while all nuclear encoded OXPHOS genes are generally upregulated in patient muscle compared to control muscle (Supplementary Fig. 3c). As TEFM is required for POLRMT processivity during the transcription elongation stage[12,13], we further investigated the effect of TEFM variants on mtDNA-encoded transcript steady-state levels. For this analysis, mt-rRNA reads corresponding to both strands were removed computationally. Skeletal muscle samples from P1 and 2 and fibroblast samples from P2, 3 and 6 showed a more severe decrease of promoter-distal mt-mRNA, consistent with the reduced levels of TEFM in patient samples (Fig. 5a, b and d, e; Supplementary Fig. 4a, b and Supplementary Fig. 5a–f). The approach used for RNA-Seq library preparation did not allow for detection of free mt-tRNAs, however, in the conditions used mt-tRNA coding regions could be captured as a part of unprocessed transcripts[24]. Consequently, by specifically analyzing the mt-tRNA regions, we could assess the effect of the TEFM depletion on the steady-state of mt-tRNA precursors. While promoter-proximal mtRNA processing intermediates are upregulated in the patient samples, there is a strong decrease of intermediates more distal from the promoter, again confirming the deleterious effect on transcription elongation in absence of TEFM protein (Fig. 5c, f, Supplementary Fig. 4c and Supplementary Fig. 5c, f). The effects of *TEFM* inactivation have been previously studied in a mouse model[14]. Loss of TEFM in mouse heart led to a similar pattern of the steady-state levels of mt-tRNA as observed in skeletal muscle of P1 and P2 (Fig. 5c and Supplementary Fig. 4c), with the most promoter proximal mt-tRNAs i.e. mt-tRNA^Phe (the closed to HSP) and mt-tRNA^Pro (close to LSP), being upregulated

(likely due to an increase of transcription initiation events) and the rest on the mt-tRNAs being downregulated. Loss of TEFM in mouse heart also showed close to normal 12S rRNA levels, but a strong reduction of the more promoter-distal 16S rRNA and reduced levels of proteins from the large mitoribosomal subunit[24]. In order to investigate if a similar effect could be detected in the patients, we analyzed rRNA levels in the patient fibroblast cells by RNA-Seq, without performing the rRNA depletion step (Supplementary Fig. 6a). We detected the reduction of both mt-rRNAs in the patient samples compared to healthy controls, with the reduction being much greater for 16S rRNA, consistent with the observations in murine heart. Despite the lower mt-rRNA levels in patients' fibroblasts, mRNA levels encoding protein components of the mitochondrial ribosomes were unchanged or slightly upregulated (Supplementary Fig. 6b), similar to the unchanged levels of mitochondrial ribosomal proteins shown by western blot (Supplementary Fig. 6c). Taken together, our results show that the *TEFM* variants lead to reduced levels of promoter-distal transcripts, likely due to impaired transcription processivity.

## In vitro transcription processivity activity of mutant TEFM enzymes

In order to provide further evidence for the pathogenicity of identified *TEFM* variants, we set out to study how the presence of the missense substitutions or the deletion, p.E162_R163del (Δ162-163), variant affect the ability of the enzyme to regulate POLRMT processivity. Human TEFM (residues 36 to 360, lacking the mitochondrial targeting sequence) and mutant variants were purified in recombinant form (Fig. 6a, Supplementary Fig. 7a). We first investigated if disease-causing mutations affect the stimulatory effect on POLRMT-driven transcription elongation. To this end, we performed run-off transcription using a linearized template containing LSP, in the presence of the different TEFM variants (Supplementary Fig. 7b). We found that the processivity of POLRMT was affected by approx. 40% for the p.(Pro157Ala), p.(Ile159Lys) and (Δ162−163) mutants and by approx. 60% for the p.(Lys188Arg) TEFM mutants, as compared to wild-type (Supplementary Fig. 7c). We next monitored the ability of wild-type TEFM and mutant derivatives to stimulate the processivity of POLRMT transcription driven from a supercoiled template containing LSP (Supplementary Fig. 7d). In this assay, the processivity of POLRMT was affected by approx. 50% for the p.(Pro157Ala), p.(Ile159Lys) and p.(Lys188Arg) mutants, while only approx. 20% reduction was observed for the (Δ162-163) TEFM mutant, as compared to wild-type (Supplementary Fig. 7e). The previous studies have shown that stimulation of POLRMT processivity by TEFM in vitro prevents the formation of G-quadraplexes at CSB II, helping the elongation complex to advance through this site[13,17]. Therefore, we also tested the ability of TEFM mutants to prevent premature transcription termination at the CSB II region. To this end we performed in vitro transcription from linearized, LSP-containing templates harbouring three different CSB II sequences characterized by varying (G)-tract lengths: G5AG7, G6AG7 and G8AG8 (Fig. 6b). These

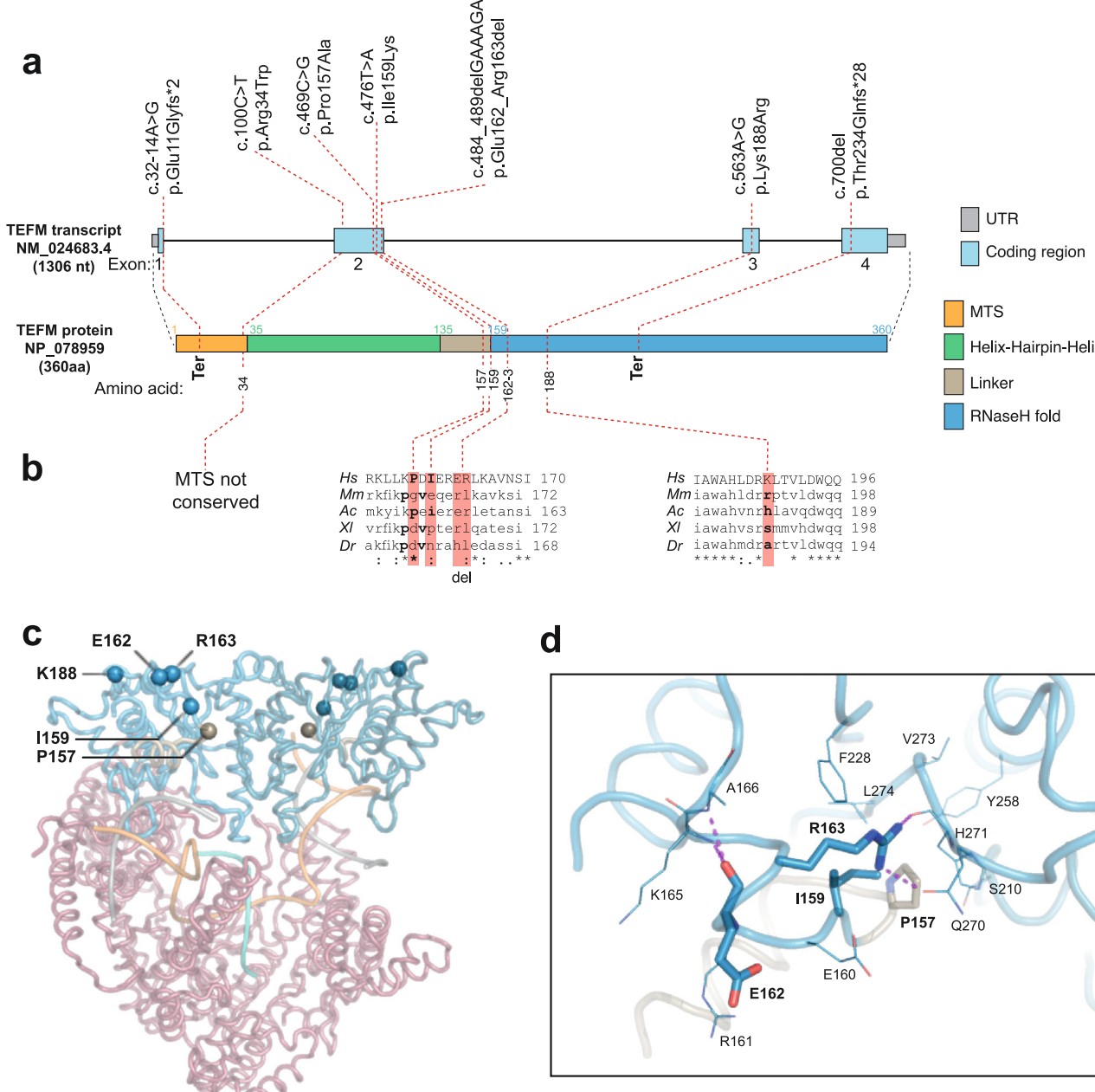

**Fig. 3 | TEFM mutations and gene/protein structure. a** Gene structure of *TEFM* with known protein domains of the gene product and localization of amino acid residues and splice site affected by mutations. The MTS (mitochondrial targeting sequence), the inter-domain linker and the two conserved fold domains (RuvC-type RnaseH-fold domain and a helix–hairpin–helix motif) are indicated. **b** Conservation of human TEFM amino acid residues affected by mutations across *Homo sapiens* (*Hs*), *Mus musculus* (*Mm*), *Anolis carolinensis* (*Ac*), *Xenopus laevis* (*Xl*) and *Danio rerio* (*Dr*). **c** The structure of the human linker (grey) and Rnase H-like domain (blue) of TEFM (PDB: 5OLA) is shown in cartoon indicating the mutation sites. POLRMT is shown in red. **d** The detailed view of the structural elements at the mutation sites.

sequence variants occur naturally in the human population and previous studies have shown that increased (G)-tract length causes higher levels of pre-termination at CSB II[25].

Consistent with the previously published data[8], we found that mutant TEFM-variants had a reduced ability to prevent premature termination at CSB II, particularly with templates containing longer (G)-tract sequences (Fig. 6c–e). Given that the composition of CSB II could have a confounding effect, we analyzed the CSBII sequences in the patients and some of their family members (Supplementary Table 4). However, we did not find any substantial differences in the 5′ part of CSB II sequences, suggesting that the sequence of the CSB II G track does not influence the molecular outcomes in the analyzed

patients. Taken together, our results show that the detected *TEFM* variants impair the activity of the enzyme, reducing the processivity of POLRMT-driven transcription.

### Early movement behaviours and phenotype of tefm-MO zebrafish

Patients 1 and 2 presented with clear neuromuscular transmission defects that were responsive to Salbutamol; a drug used to treat patients with a primary neuromuscular transmission impairment[26]. To investigate whether loss of TEFM is associated with impaired neuromuscular function in vivo, we generated a *tefm* knockdown zebrafish model using a morpholino antisense oligonucleotide (MO). Success of

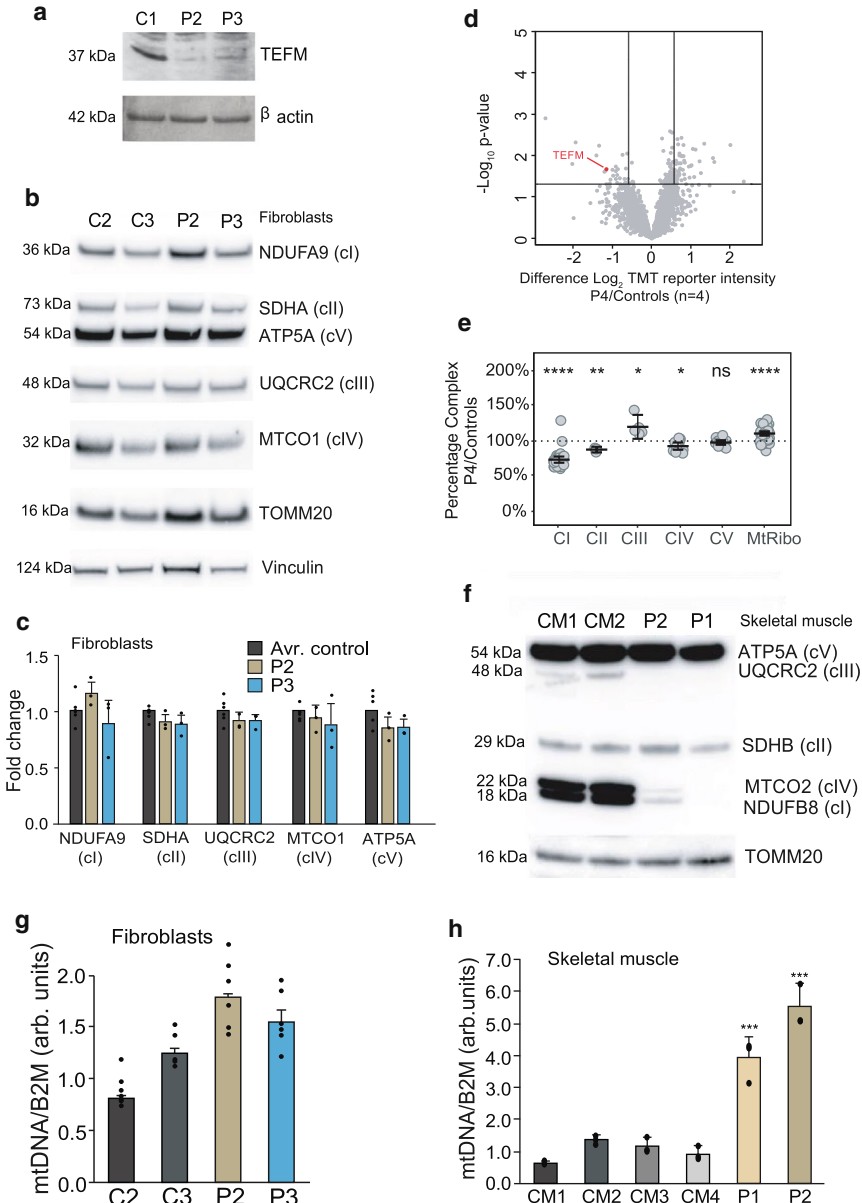

**Fig. 4 | The levels of TEFM and OXPHOS components and mtDNA copy number in patient samples. a** Western blot detecting the TEFM protein in human dermal fibroblasts from patients 2 (P2) and 3 (P3). β-actin was used as a loading control. **b** Representative example of western blot analysis of NDUFB9 (complex I, cI), SDHA (complex II, cII), UQCRC2 (complex III, cIII), MTCO1 (complex IV, cIV), and ATP5A (complex V, cV) for fibroblasts from patients 2 (P2) and 3 (P3) and wild-type controls (C2 and C3). TOMM20 and Vinculin were used as loading controls. **c** Quantification of 3 western blot experiments as per (b). Data were statistically analyzed by two-tailed Student's t-test. Error bars = mean ± 1 standard deviation, n = 3. **d** Volcano plot comparing whole-cell protein abundances for proteins quantified in P4 fibroblasts (n = 2 technical) relative to control fibroblasts (n = 4 biological). TEFM protein is highlighted in red. Significance lines were plotted at p-value <0.05 and log₂ difference 0.585 (1.5-fold change equivalent) from two-sample t-test analysis. **e** Relative complex abundance of quantitative proteomics data plotted as a ratio of each subunit in TEFM/controls and represented as a percentage. The middle bar represents the mean value for the subunits in each

complex, upper and lower bars represent 95% confidence interval of the mean value. A paired t-test calculated the significance between the control and patient for each complex. P values are indicated as ****p ≤ 0.0001; **p ≤ 0.01, *p ≤ 0.05, ns = not significant. Middle dotted line represents 100%, or no change between the TEFM and controls. **f** Representative example of western blot analysis of NDUFB8 (complex I, cI), SDHB (complex II, cII), UQCRC2 (complex III, cIII), MTCO2 (complex IV, cIV), and ATP5A (complex V, cV) for skeletal muscle samples from patients 1 (P1) and 2 (P2) and wild-type controls (CM1 and CM2). **g** mtDNA copy-number determination by qPCR of mtDNA fragments relative to the nuclear *B2M* gene for fibroblasts of P2 and P3 and the wild-type controls. Statistical analysis was carried out using a two-tailed student's t-test. Error bars = mean ± 1 standard deviation, n = 6 (technical replicates, ns ≥ 0.05). **h** mtDNA copy-number determination by qPCR of mtDNA fragments relative to the nuclear *B2M* gene in skeletal muscle samples of P1 and P2 and wild-type controls. Statistical analysis was carried out using a two-tailed student's t-test. Error bars = mean ± 1 standard deviation, n = 3 (technical replicates); *p ≤ 0.05; **p ≤ 0.01. Source data are provided as a Source Data file.

the *tefm* knockdown was demonstrated by RT-PCR (Supplementary Fig. 8a). Survival of *tefm*-MO fish was not significantly affected during the first 2 days of life (Supplementary Fig. 8b). Head angle and length, both indicators of developmental stage[27], were also unaffected by *tefm* knockdown (Supplementary Fig. 8c–e).

Neuromuscular development within the tail musculature in zebrafish occurs during the first few days of life, starting with the outgrowth of primary motor neurons from the spinal cord at 18 h post fertilization. This event coincides with the onset of spontaneous muscle twitches of the zebrafish from within the chorion

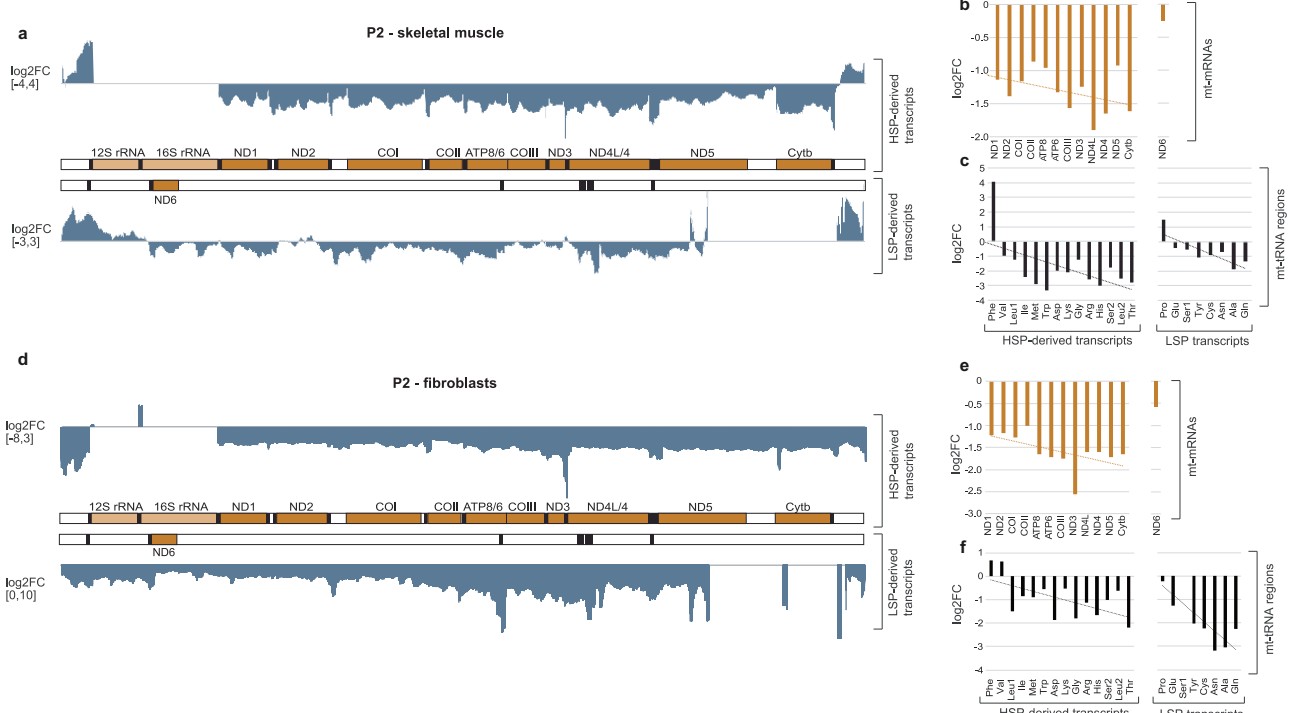

**Fig. 5 | Mitochondrial transcriptome in TEFM patients. a** Transcriptome-wide RNA-Seq analysis showing the effects of TEFM loss in skeletal muscle of patient 2 (P2) on mtRNA transcripts. A map of mtDNA showing the change in sequence read coverage ($log_2$ fold change between patient 2 and two healthy controls) on both strands. **b** Quantification of the difference in reads per gene between skeletal muscle from P2 and from two healthy controls ($n = 3$ technical replicates) for mt-mRNA derived reads for both strands, from proximal to distal from the promoter and **c** reads corresponding to mt-tRNA for both strands in the order they are transcribed ($n = 3$ technical replicates) **d** Transcriptome-wide RNA-Seq analysis of the effects of TEFM loss in fibroblast cells of P2 on mtRNA transcripts. A map of mtDNA showing the change in sequence read coverage (log2 fold change between P2 and fibroblast cells from healthy controls) on both strands. **e** Quantification of the differences in number of reads per gene between fibroblasts from P2 and from two healthy controls ($n = 3$ technical replicates) for mt-mRNA reads and **f** reads corresponding to mt-tRNA fragments in the order they are transcribed from the promoter ($n = 3$ technical replicates). All trend lines were fitted by linear regression. Source data are provided as a Source Data file.

(egg sac)[28]. Muscle twitches or 'chorion rotations' are not dependent on supraspinal input and are important for proper development of the neuromuscular system in the zebrafish[28]. At 1 day post fertilization (dpf), *tefm*-MO fish performed approximately 50% fewer chorion rotations per minute than control-MO injected fish (Fig. 7a). Mean chorion rotation duration was significantly increased in *tefm*-MO fish, from an average of 0.054 s in controls to 0.059 s (Fig. 7b). The percentage of time spent moving during 1 min was significantly reduced in *tefm*-MO fish, from an average of 0.54−0.38% (Fig. 7c).

### Neuromuscular junction morphology in tefm-MO zebrafish

Neuromuscular junction morphology can also be studied in zebrafish, helped by their easily accessible muscle tissue and transparent skin during early development. At 2 dpf whole fish were stained for SV2 to detect motor neurons and a fluorophore-conjugated α-bungarotoxin to detect AChRs on muscle fibres (Fig. 7d). Presence of gross morphological defects in *tefm*-MO fish motor neuron development can be observed (white arrows, Fig. 7d), including instances of improper migration patterns or absence of neuron outgrowth. Quantification of presynaptic features revealed a significant reduction in the average size of SV2-positive clusters in the tail musculature (Fig. 7e). The number of SV2-positive clusters per 100 μm² was not significantly different in *tefm*-MO fish as compared to controls (Fig. 7f), nor was the average size of AChR clusters or number per 100 μm² (Fig. 7g/h). Co-localization analysis revealed a significant reduction in the amount of SV2 co-occurring with α-bungarotoxin and α-bungarotoxin with SV2 on both fast muscle and slow muscle (at myosepta) in *tefm*-MO fish as

compared to WT (Fig. 7i−l). In summary, these experiments suggest that neuromuscular junction development in zebrafish embryos is dependent on *tefm*.

### Expression of OXPHOS components in tefm-MO zebrafish

To determine whether *tefm*-MO zebrafish exhibit changes in expression of mitochondrial OXPHOS component expression, 2 dpf fish were probed with antibodies against proteins within complex I, IV and V. There were no significant differences in expression of NDUFA9 (complex I) or ATP5a (complex V, Supplementary Fig. 9a, b). MTCO1 (complex IV) shows decreased expression in *tefm*-MO fish as compared to control MO fish (Supplementary Fig. 9c).

### F0 knockout of zebrafish-tefm using CRISPR/Cas9 mutagenesis

Targeting exon 2 and 3 of zebrafish *tefm*, three different Cas9/guide RNA (gRNA) ribonucoprotein (RNPs) complexes were co-injected into newly fertilized wild-type zebrafish embryos to generate a *tefm*-knockout in the F0 generation. High resolution melt analysis (HMA) indicated that gRNAs 1 and 2 had effectively induced multiple insertions and deletions (indels) into both exon 2 and 3 of zebrafish *tefm*, indicated by the presence of multiple high weight bands in the injected samples (Supplementary Fig. 10b). RT-qPCR of mitochondrially encoded mRNAs shows a significant reduction in some HSP-derived transcripts in the *tefm* (F0) injected relative to the uninjected controls (Supplementary Fig. 10c), showing some similarity to what we see in the patient fibroblast and muscle samples. There was no change in *nd6* expression, the only LSP-derived mitochondrial mRNA transcript. These results indicate that mutations induced in zebrafish

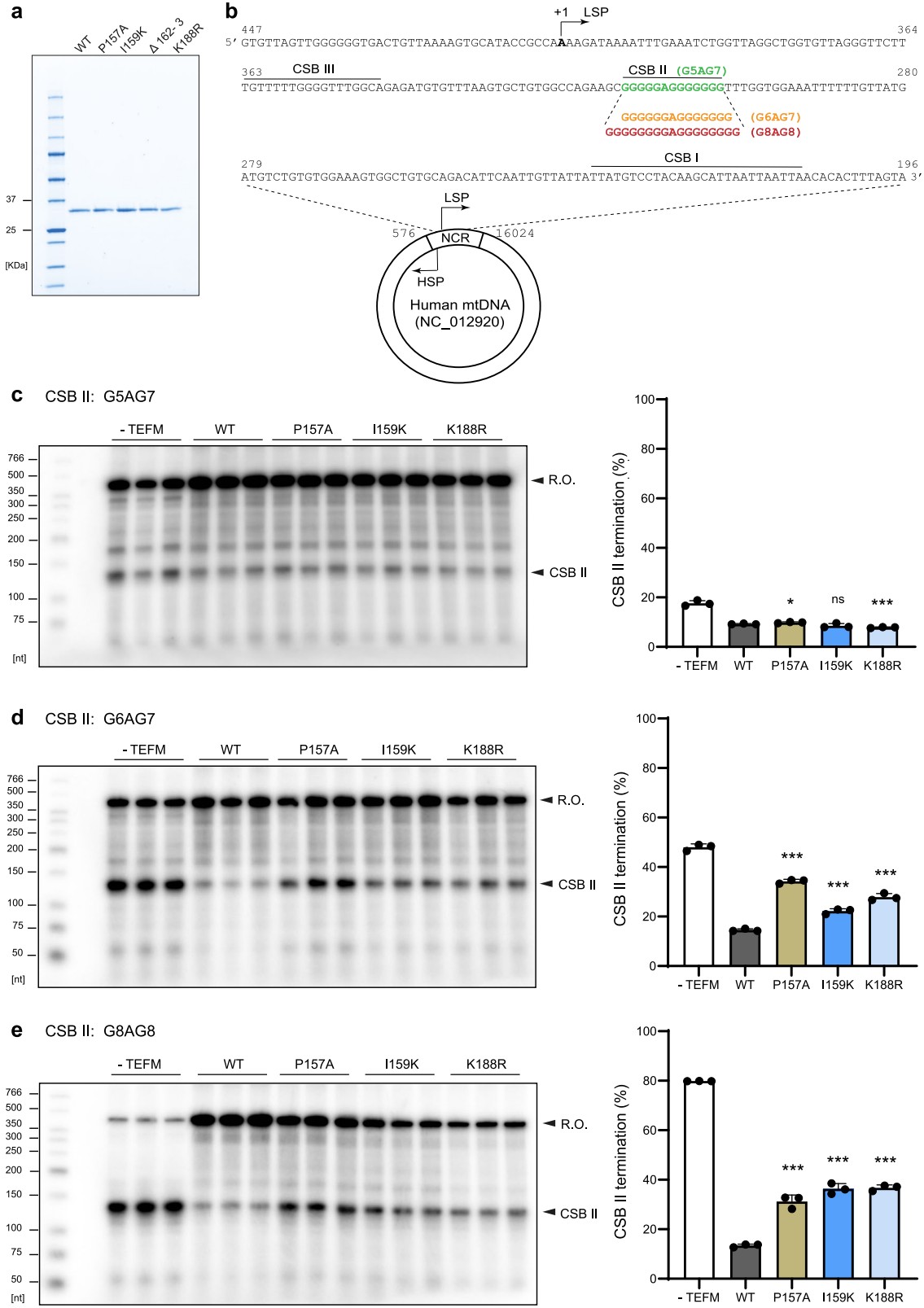

*tefm* have a significant reduction on HSP-derived mitochondrial mRNA transcripts.

**Spontaneous movement in tefm (F0) knockout zebrafish**
Spontaneous movement in zebrafish varies when larvae are exposed to light or darkness[29]. Five dpf injected – *tefm* (F0) zebrafish have a significantly lower average total distance moved when light was off

compared to uninjected controls (Fig. 8a, b). Reduction in movement in the *tefm* (F0) when the light was off, become more pronounced as time progressed. There was no significant difference in movement when light was on. In summary, mutations in *tefm* of 5 dpf zebrafish impact darkness induced spontaneous movement compared to uninjected controls and this shows some fatigability with time confirming the phenotype observed in the *tefm*-MO injected fish.

**Fig. 6 | In vitro analysis of TEFM variants. a** SDS-PAGE analysis of the purified recombinant WT TEFM (amino acids 36–360) and the TEFM mutants used in the in vitro experiments. Molecular weight marker is indicated in the first lane. **b** DNA sequence (447–196) included in the mitochondrial Non-Coding Region (NCR), harbouring the transcription start site (+1) of the light strand promoter (LSP) and the Conserved Sequence Blocks (CSB) I, II and III. Highlighted are the CSB II sequence variants used in biochemical assays, G5AG7, G6AG7 and G8AG8. **c–e** CSB II sequence heterogeneity affects the capacity of TEFM mutants of preventing premature transcription termination at the CSB II region. Left panels: in vitro transcription from linearized templates that generate transcripts of ~400 nt (R.O.). Three templates with different (G)-tract sequence at CSB II were used: G5AG7 (**c**), G6AG7 (**d**) and G8AG8 (**e**). WT and TEFM mutants (250 fmol as a dimer) were assayed for their ability to abolish transcription termination at CSB II in each template. Right panels: The mean value of termination at different CSB II variants in the in vitro transcription. Error bars represent mean ± SD of three technical repeats, ns not significant, *$p \leq 0.05$; **$p \leq 0.01$; ***$p \leq 0.001$ (two-tailed Student's *t* test for comparison to reactions with WT TEFM). Source data are provided as a Source Data file.

## Discussion

The mitochondrial genome encodes essential subunits of the OXPHOS system and RNA components necessary for intra-mitochondrial translation, whereas the nuclear genes code for the proteins responsible for mtDNA transcription, post-transcriptional RNA processing and translation. In recent years there has been a rapid development in our understanding of these machineries both in human health and in disease states. Dysfunction of mitochondrial gene expression, caused by mutations in either the mitochondrial or nuclear genomes, is associated with a diverse group of human disorders characterized by impaired mitochondrial respiration. Within this group, an increasing number of mutations have been identified in nuclear genes involved in mtRNA metabolism[1,19,23,30].

Patients with *TEFM* variants present with a wide range of clinical presentations including neonatal lactic acidosis, epileptic encephalopathy, developmental delay and intellectual disability with nonspecific changes on brain MRI or mitochondrial myopathy with a treatable neuromuscular transmission defect. Human *TEFM* variants in our cohort all show relatively high residual activity (hypomorphic alleles) and there is a lack of homozygous LoF variants in gnomAD. Intercrossing of heterozygous mice produced no viable homozygous knockout (*Tefm* −/−) mice, demonstrating that loss of Tefm results in embryonic lethality[14]. However, very low residual TEFM protein level in skeletal muscle of P1 and P2 was compatible with life. The variable severity and phenotypes associated with TEFM deficiency in humans may be due to differences in residual activity of the mutant enzyme or could be influenced by other factors affecting mitochondrial transcription and respiratory chain activity.

Autosomal recessive and dominant variants have been recently identified in the *POLRMT* gene in eight individuals from seven unrelated families[19]. The clinical presentation of these patients resembles *TEFM* mutations with global developmental delay, hypotonia, short stature, intellectual disability, dysmorphic features and/or epileptic seizures in childhood, while other subjects developed muscle weakness and atrophy or autosomal dominant progressive external ophthalmoplegia. Brain MRI showed high signal intensity in the subcortical white matter in some patients, while volume loss of the white matter with ventriculomegaly and thinning of the corpus callosum in others, and some patients with predominant muscle involvement had normal brain MRI. Similarly, the clinical and brain abnormalities were diverse in our cohort of patients with *TEFM* variants, suggesting that mitochondrial transcription and its elongation are crucial for a wide range of neurons and for skeletal muscle. Diseases caused by *POLRMT* and *TEFM* mutations are similar and a wide variety of features overlap with other mitochondrial diseases, establishing defective mitochondrial transcription elongation as an important disease mechanism.

Some TEFM patients in our study show a prominent neuromuscular junction (NMJ) defect responsive to treatment, which has a positive impact on mobility and quality of life. The responsiveness of these patients to salbutamol, a drug commonly used to alleviate symptoms of fatigable muscle weakness in the Congenital Myasthenic Syndromes (primary disorders of the NMJ), suggests an involvement of TEFM in neuromuscular structure or function. To investigate this, we generated a zebrafish model of TEFM deficiency using morpholino antisense oligonucleotide technology to knockdown expression of *tefm* during the first few days of development. The knockdown is mosaic, allowing the zebrafish to progress through development without experiencing severe, universal deficits in RC activity.

Fish with reduced expression of *tefm* moved significantly less over the period of 1 min, as found in other zebrafish models of neuromuscular defects[31,32], and the duration of movements was longer, suggesting a slower twisting motion. Reduced movement occurs naturally as zebrafish move from 19 h post fertilization (hpf) towards 27 hpf, or if development is delayed and movements have not yet initiated. However, we assessed the length and head angle of these fish to ensure they are at the same developmental time point and no differences were found[33]. In addition, a CRISPR/Cas9 F0 *tefm* knockout model also displayed defects in movement at 5 dpf, with defects becoming more pronounced over time of the assay, which may reflect some involvement of the NMJ. Further evidence towards a role for *tefm* in zebrafish NMJ development is provided by the decrease in co-occurrence of presynaptic and post-synaptic NMJ components on fast and slow muscle and a reduction in average pre-synaptic cluster size in *tefm*-MO fish. As the NMJ has a high concentration of mitochondria and requires energy for formation, vesicle release/recycling and assembly of the presynaptic cytoskeleton[34], it is likely that the neuromuscular synapse is highly vulnerable to deficiencies in mitochondrial function. Alternatively, TEFM may have a role at the NMJ beyond promoting transcription elongation in mitochondria, rendering it important for proper synapse formation, however, further research will be required to address this. The detection of the presynaptic NMJ defect in *tefm*-MO treated zebrafish supports the link between *TEFM* variants and NMJ dysfunction and highlights that improving neuromuscular transmission may improve fatigability in patients carrying biallelic variants in the *TEFM* gene. Many mitochondrial disorders affect the brain, nerve and muscle, but neuromuscular transmission disturbances caused by mitochondrial defects have not been appreciated until very recently[35]. In a cohort of 80 patients with various genetic forms of mitochondrial disease, detailed clinical and neurophysiologic testing including single-fibre electromyography detected neuromuscular transmission defects in 26%. The highest prevalence was in patients with pathogenic dominant *RRM2B* variants (50%), but abnormalities were found in a wide range of mitochondrial genotypes, including common mtDNA variants. The presence of NMJ abnormalities was strongly associated with coexistent myopathy, however 15% of patients with NMJ abnormalities had no evidence of either myopathy or neuropathy[35]. Whether modulators of neuromuscular transmission have a role in treating these patients has not been studied systematically.

Despite the increased utility of genetic testing and variant pathogenicity prediction tools, providing proof of pathogenicity of novel variants remains challenging. Follow up functional studies in vitro should, therefore, be included as an integral part of the evaluation[36]. Our multifaceted approach involving transcriptomics, quantitative proteomics, activity analysis of the recombinant TEFM protein and the assessment of the protein stability in cultured patient fibroblasts provided convincing pathogenicity evidence for the missense and deletion variants identified in P1–4 and P6–P7. It is unlikely that a single mutation can affect the dimerization of TEFM as the

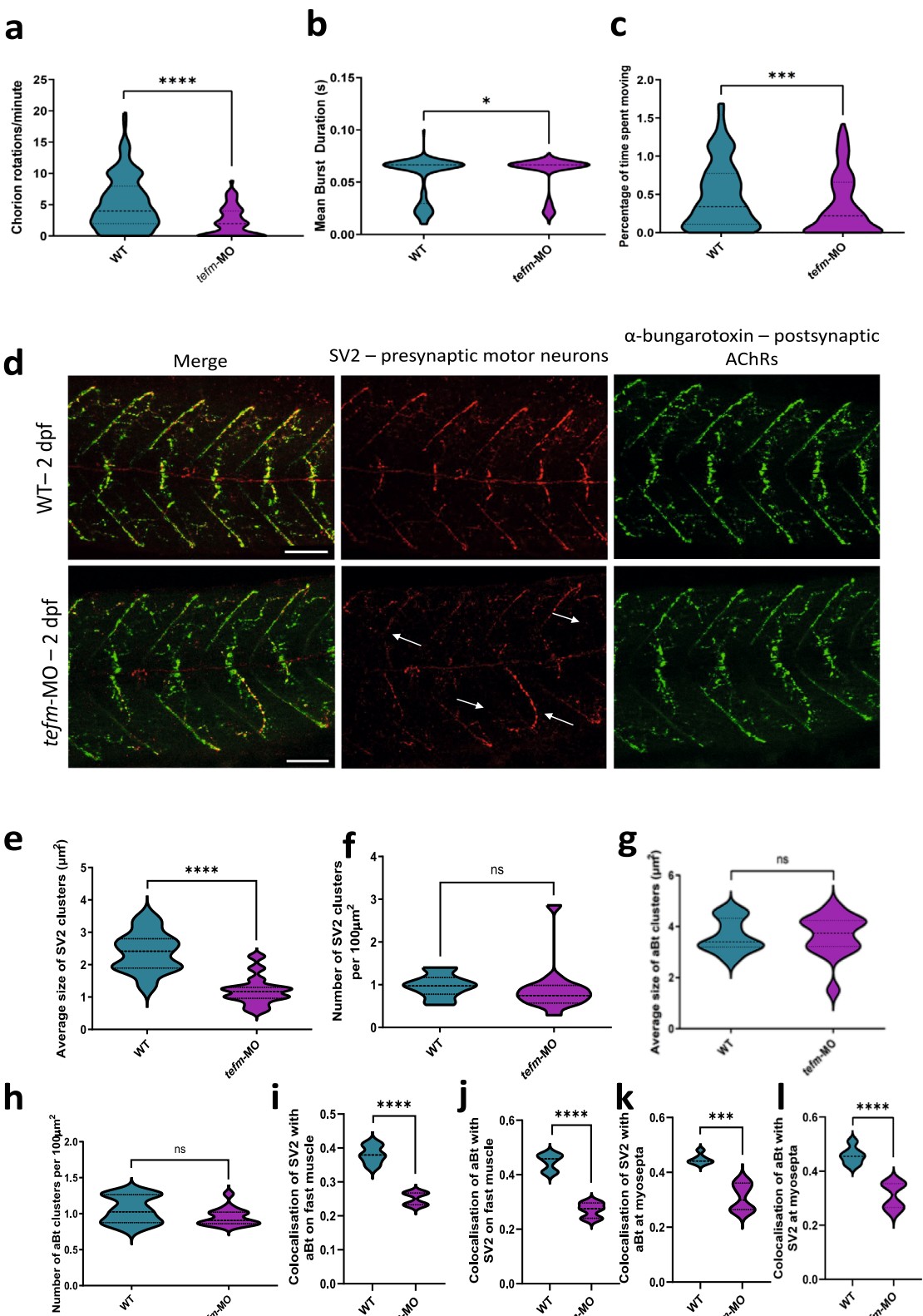

identified mutations are far from the dimer interface formed by the long alpha-helices in the C-terminus of TEFM. Most mutations cluster in an important structural element of TEFM - the linker subdomain, which has been previously identified as critical to TFEM function, and mutations there dramatically affect transcription[22]. However, the pathogenicity of the missense variants found in P5 remain somewhat unclear. Owing to the unavailability of P5 material, we could not test

the effect of p.(Lys188Arg) and p.(Arg34Trp) on mitochondrial transcriptome, proteome or TEFM steady-state level. Both variants found in P5 have relatively high population allele frequency. While p.(Lys188Arg) had a clear detrimental effect on recombinant TEFM activity, we were unable to study the effect of the p.(Arg34Trp) change in the in vitro transcription assay, as it resides in the predicted mitochondrial targeting sequence (MTS) (Fig. 3), which needs to be

**Fig. 7 | Early movement behaviours and neuromuscular junction morphology of tefm-MO and tefm (F0) zebrafish. a** Chorion rotations per minute ($p < 0.0001$, two-sided Mann–Whitney test, $n = 133$ control and 132 *tefm*-MO), ****$p < 0.0001$. **b** Mean chorion rotation duration in seconds ($p = 0.015$, two-sided Mann–Whitney test, $n = 128$ control and 121 *tefm*-MO), *$p < 0.05$. **c** Percentage of time spent moving in control and *tefm*-MO zebrafish at 1 dpf ($p = 0.0006$, two-sided Mann–Whitney test, $n = 133$ control and 167 *tefm*-MO), ***$p < 0.001$. **d** Representative images of neuromuscular junctions from wild-type (WT) and tefm-MO zebrafish at 2 dpf (days post fertilization). Acetylcholine receptors (AChRs) are stained with fluorophore bound α-bungarotoxin (aBt, green), and motor neurons detected with an antibody against synaptic vesicle protein 2 (SV2, red). White arrows indicate points of interest, such as lack of neuron outgrowth or improper neuron migration. Scale bar = 50 μm. **e** Average size of SV2 clusters (p < 0.0001, two-sided unpaired *t*-test, $n = 12$ WT and $n = 15$ *tefm*-MO), ****$p < 0.0001$. **f** Number of SV2-positive

clusters per 100 μm² ($p = 0.124$, two-sided Mann–Whitney test, $n = 12$ WT and $n = 15$ *tefm*-MO), ns not significant. **g** Average size of aBt clusters (p = 0.788 two-sided Mann–Whitney test, $n = 12$ WT and $n = 15$ *tefm*-MO), ns not significant. **h** Number of aBt-positive clusters per 100 μm² ($p = 0.196$ two-sided Mann–Whitney test, $n = 12$ WT and $n = 15$ *tefm*-MO), ns not significant. **i** Colocalization of SV2-positive signal with αBT on fast muscle using Mander's correlation coefficient (0 = no colocalization, 1 = full colocalization, $p < 0.0001$, two-sided nested *t*-test, $n = 10$ WT and $n = 15$ *tefm*-MO), ****$p < 0.0001$. **j** Colocalization of αBT with SV2-positive signal on fast muscle ($p < 0.0001$, two-sided nested *t*-test, $n = 10$ WT and $n = 15$ *tefm*-MO), ****$p < 0.0001$. **k** Colocalization of SV2-positive signal with αBT on slow muscle ($p = 0.0003$, respectively, two-sided unpaired *t*-test, $n = 10$ WT and $n = 15$ *tefm*-MO), ***$p < 0.001$. **l** Colocalization of αBT with SV2-positive signal on slow muscle ($p < 0.0001$, respectively, two-sided unpaired *t*-test, $n = 10$ WT and $n = 15$ *tefm*-MO), ****$p < 0.001$. Dashed line shows the median, dotted lines show the quartiles.

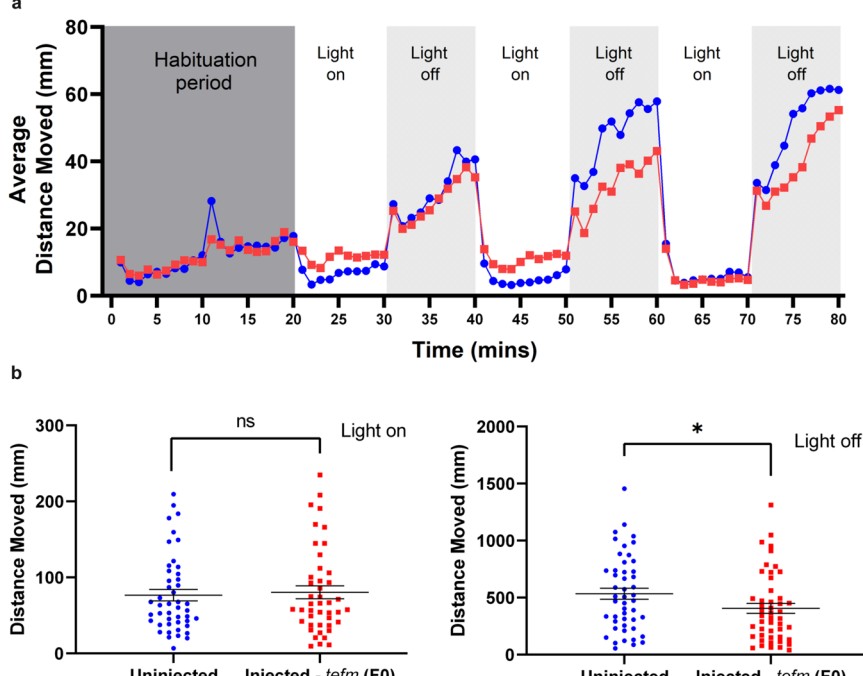

**Fig. 8 | Movement analysis of tefm (F0) zebrafish. a** Average distance moved (mm) of 5 dpf uninjected (blue circles) and injected - *tefm* (F0) (red squares) zebrafish exposed to three cycles of 10 min light on and 10 min light off after a 20-min habituation period. Each point represents 1 min, $n = 48$ uninjected and $n = 48$ injected - *tefm* (F0). **b** Total distance moved (mm) of 5 dpf uninjected (blue circles) and injected - *tefm* (F0) (red squares) zebrafish for the first 5-min of light on (left image) and light off (right image) periods from (**a**). Right image (light off), $p = 0.0395$, two-sided Mann–Whitney test, $n = 48$ uninjected and 48 injected − *tefm* (F0). Left image (light on), $p = 0.5767$, two-sided Mann–Whitney test. Error bars = mean ± SEM ns not significant, *$p < 0.005$. Source data are provided as a Source Data file.

removed to allow the efficient production of the recombinant protein. The predicted length of the MTS in TEFM is 36 aa. The mitochondrial processing peptidase (MPP) recognizes a large variety of MTSs in mitochondrial preproteins and cleaves a single site, often including arginine, at the −2 position[37,38]. Therefore, it is possible that Arg34 is the likely recognition site for the MPP.

The predicted length of the MTS in TEFM is 36 aa, it is possible that Arg34, which is the likely recognition site for the mitochondrial processing peptidase (MPP), is absolutely conserved in mammalian TEFM. MPP recognizes a large variety of MTSs in mitochondrial preproteins and cleaves the single site, often including arginine, at the −2 position[37,38]. Therefore, it is expected that the p.(Arg34Trp) mutation found in P5 will result in a lack of TEFM processing in mitochondria and a partially unfolded and partially active protein.

Transcripts originating from LSP, which are prematurely terminated at CBSII have been suggested to act as primers for H-strand mtDNA replication[15,16]. Owing to the capability of TEFM to affect

premature transcription termination at CSBII, it has been proposed to act as the switch between replication and transcription. According to this model, enhanced transcription termination at CSBII would yield more RNA primer resulting in more mtDNA replication, whereas reduced CSBII termination events would lead to higher readthrough and transcription rates (also see "Introduction")[17]. TEFM normally stimulates transcription to pass through CSBII without termination. Here, we showed that the p.(Pro157Ala) TEFM mutant, identified in P1 and P2, had a reduced ability to prevent premature termination at CSB II, particularly with templates containing longer (G)-tract sequences (Supplementary Fig. 7c–e). Furthermore, we observed in P1 and P2 skeletal muscle samples that mtDNA copy number was higher as compared to controls (Fig. 4g). These two results could potentially support the TEFM replication-transcription switch model. However, in vivo data obtained in the TEFM mouse knockout showed that the lack of TEFM did not result in upregulation of mtDNA replication initiation and the lack of TEFM caused a decrease in de novo mtDNA

replication[14]. The observed upregulation of mtDNA copy number in TEFM knockout mice was attributed to the compensatory increase of mitochondrial biogenesis, typically occurring in severely respiratory chain deficient tissues[39]. P1 and P2 skeletal muscle samples also exhibited respiratory chain deficiency (Fig. 4f) that could trigger mitochondrial biogenesis and provide an alternative explanation for the elevated mtDNA in this tissue. On the other hand, no increase in TOMM20 protein was observed, suggesting that mitochondrial mass was not changed in P1 and P2 skeletal muscle. More research would be required to test the TEFM-related replication-transcription switch model.

In summary, we report the identification and characterisation of pathogenic variants in the mitochondrial transcription elongation factor, TEFM, that underlie the mitochondrial disease-related presentations in seven patients from five families. Our in vivo and in vitro functional studies establish TEFM as a candidate disease gene that should be considered in the genetic diagnostic workup of patients that present with neonatal or childhood onset mitochondrial encephalomyopathy, fatigable muscle weakness, epilepsy and variable degree of intellectual disability. Neuromuscular transmission studies may reveal abnormal neuromuscular transmission in patients with fatigable muscle weakness carrying causative *TEFM* variants, which can benefit from Salbutamol therapy. Together with the recent report of several pathogenic variants in the mitochondrial RNA polymerase, POLRMT, our study indicates that defects in the human mitochondrial transcription machinery are more prevalent than previously thought.

## Methods

### Ethics statement
Informed consent for diagnostic and research-based studies was obtained for all subjects in accordance with the Declaration of Helsinki protocols and approved by local institutional review boards (Yorkshire & The Humber - Leeds Bradford Research Ethics Committee (13/YH/0310), the Sydney Children's Hospitals Network Human Research Ethics Committee (HREC/10/CHW/113) and the Royal Children's Hospital (Melbourne) Human Research Ethics Committee (HREC 34228), Federico II University Hospital Ethics Committee (48/16). The clinicians in India, France, Australia, Italy and Egypt confirmed in each country that the patients consented to the research and to the publication of this research on their own consent forms approved by local ethics committees on their local language.

All zebrafish experiments carried out in Canada were performed in accordance with guidelines from the Canadian Council on Animal Care and were approved by the University of Ottawa Animal Care Committee. All zebrafish experiments performed in the UK were carried out in accordance with UK Home Office Guidelines, UK Animals (Scientific Procedures) Act 1986 and with approval of the local University of Cambridge ethics committee. Adult UK zebrafish were maintained under project licence P1CG6E735.

### Patient recruitment and sample collection
Patients presented with a history of childhood-onset muscle weakness and/or developmental delay and underwent detailed clinical examination. DNA, tissues including muscle and skin fibroblasts, and medical records were obtained based on standard procedures. Patients 1 and 2 were first identified in RD-Connect GPAP (Genome-Phenome Analysis Platform)[40], Patients 3–5 were identified through GeneMatcher[41], while patients 6 and 7 were matched within the RD-Connect Genome-Phenome Analysis Platform (GPAP).

### Exome and genome sequencing, variant prioritization, re-evaluation and verification
Trio WES was performed in families 1, 2 and 4, while trio WGS was done in family 3 and proband only WES in family 5.

Variants were confirmed by Sanger sequencing. Trio WES in P1 and P2 was performed at the Broad Institute (Boston, USA) using Standard Germline Exome version 5, handled by the Genomics Platform's Core Exome product team. The process includes sample preparation (Illumina Nextera), hybrid capture (Illumina Rapid Capture Enrichment 37 Mb target), sequencing (Illumina, 150 bp paired reads) and identification quality control check. The hybrid selection libraries typically meet or exceed 85% of targets at 20x, comparable to ~55x mean coverage. Copy number variants in the whole exome sequencing data were detected using the Genome Analysis Toolkit (GATK)'s germline CNV caller (gCNV). Sequencing biases were modelled via negative-binomial factor analysis to reduce technical artifacts and copy number states and genomic regions of low and high copy number variability were modelled using hierarchical hidden Markov models (HHMM) to impute rare and common copy number variants.

WES was performed on genomic DNA (gDNA) extracted from fibroblasts of Individual 3 and from leucocytes of his parents. The exonic regions and flanking splice junctions of the genome were captured using the Twist Human Core Exome Kit (Twist Bioscience, San Francisco, CA, USA). Massively parallel (NextGen) sequencing was done on an Novaseq (Illumina, San Diego, CA, USA; NovaSeq Control Software v1.8.) with 75 bp or greater paired-end reads. Reads were aligned to human genome build GRCh38 by using the Burrows–Wheeler Aligner (BWA) tool for Individuals 1 and 2 and were analyzed for sequence variants using a custom-developed analysis tool. Average coverage was 65X (99.2% > 25X) for individual 3, 75X (99.6% > 25X) for his father and 73X (99.3% > 25X) for his mother. Ensembl's VEP (Variant Effect Predictor, release 92) was used to annotate the variant. Analysis of WES data prioritized variants based on the allele frequency (minor allele frequency [MAF] ≤ 0.1%) in population database (gnomAD, EVS and 1000 genomes) and no homozygotes in public databases, the presence in databases of medically relevant variants (ClinVar, Human Gene Mutation Database), the predicted impact on coding sequence, the mode of inheritance and the compatibility with the clinical synopsis of the OMIM database when available. After looking for de novo, X-linked or biallelic variants, *TEFM* was the only candidate gene.

For P4 Trio WGS on gDNA from blood was conducted at the Kinghorn Centre for Clinical Genomics (Garvan Institute, Sydney) as described[42]. WGS TruSeq Nano libraries were prepared and loaded onto a HiSeq X Ten sequencer (Illumina; control Software v3.0.29.0) and 2 × 150 bp paired-end sequencing was performed at the Kinghorn Centre for Clinical Genomics (KCCG). More than 75% of bases having Q30 base quality and mean coverage of >30×. Raw sequencing data were converted to FastQ format using Illumina's bcl2fastq converter (v2.15.0.4) and read quality was evaluated using FastQC. Sequences were aligned to the b37d5 human reference genome using Burrows–Wheeler Aligner (BWA, v0.7.12-r1039), coordinate-sorted using Novosort (v1.03.04, Novocraft Technologies Sdn Bhd, Selangor, Malaysia), and improved using GATK (v3.4-46-gbc02625)[43] indel realignment and base recalibration to generate BAM files. Variants were called using GATK (v3.4-46-gbc02625)[43] HaplotypeCaller followed by joint variant calling with GenotypeGVCFs and VariantRecalibration. The resulting multi-sample VCF file was annotated using ENSEMBL's Variant Effect Predictor (v74)[44] and converted to an SQLite database using gemini (v0.17.2)[45]. Gemini databases were imported into Seave[46] which was used to perform variant filtration and prioritization.

For P5 genetic studies were done by trio WES with enrichment with SureSelect Clinical Research Exome (Agilent, Technologies, Santa Clara, CA, USA) and sequencing by the NextSeq 500 sequencing system (Illumina, San Diego, CA, USA), according to previously published methods[47] and a customized method for calling, annotation, filtering and prioritization of the variants[47–50]. Libraries were prepared using the Kapa HTP kit (Illumina, San Diego, CA, USA) and capture was

performed using the SeqCap EZ MedExome (Roche NimbleGen Madison, WI, USA). Sequencing was done on an Illumina HiSeq2500 HTv4 (Illumina, San Diego, CA, USA) with paired-end 125-bp reads. Read alignment to hg19 and variant calling were done with a pipeline based on BWA-MEM0.7 and GATK 3.3.0. The median coverage of the captured target region was at least 100x. Variant annotation and prioritizing were done using Cartagenia Bench Lab NGS (Agilent Technologies). We excluded variants located outside the exons and intron/exon boundaries and variants with a minor allele frequency (MAF) of >1% in control databases, including dbSNP137 (http://www.ncbi.nlm.nih.gov/projects/SNP), 1000 Genomes Project (release of February 2012), and Exome Variant Server (EVS), NHLBI Exome Sequencing ProjectNational Heart, Lung, and Blood Institute GO Exome Sequencing Project (ESP5400 release) (http://evs.gs.washington.edu/EVS/) and our in-house exome controls. Variants that fitted with a de novo or recessive mode of inheritance were further analyzed.

For P6 and P7 genetic studies were done as described[51]. The exome was captured using the SureSelect Human All Exons v3 (20 patients), v4 (11 patients), and v5 (19 patients) reagents (Agilent Inc.® Santa Clara, CA). Sequencing was performed in an IlluminaHiSeq 2000 instrument. Each exome library was indexed, separated into two equal halves, and sequenced in two different lanes. The raw results were analyzed using a customized pipeline that utilizes published algorithms in a sequential manner (BWA for mapping the reads, SAMtools for detection of variants, Pindel for the detection of indels, and ANNOVAR for the annotation). The entire coding sequence corresponding to the human RefSeq coding genes was used as the reference for the calculation of coverage and reads on target. All experiments were performed using the manufacturer's recommended protocols without modifications.

## Complementary DNA (cDNA) studies

Cultured fibroblasts from P4 and an age-matched control were treated with 100 ng/μL cycloheximide for 24 h before RNA extraction[52]. RNA was isolated using the RNeasy Plus kit (Qiagen) following the manufacturer's protocol, cDNA was synthesized using the SuperScript III First-Strand synthesis kit (Thermo Fisher Scientific) using the manufacturer's instructions. Polymerase chain reaction (PCR) was performed on cDNA using primers designed to amplify exons 1–4 of *TEFM* (forward primer ATGAGCGGGTCTGTCCTCTT, reverse primer GATGGGAAGAACACCCGAGG) followed by Sanger sequencing.

## In silico modelling

The structure of TEFM (Protein Data Bank: 5OLA) was used to model the position and function of residues at which substitutions were observed in this study. Molecular structures were displayed using PyMOL. The overall structure is shown using cartoon representation with individual residues displayed as spheres or sticks and polar contacts as dashed lines.

## DNA isolation and mtDNA copy number

DNA was isolated from cultured fibroblasts using a DNeasy Blood and Tissue Kit (Qiagen) and quantified using a nanodrop 2000 spectrophotometer (ThermoFisher). Muscle samples were homogenized with a TissueRuptor II (Qiagen) prior to isolation and quantification via the same method. All samples were diluted to 2 ng/μL with nuclease free water. MtDNA copy number was quantified using a duplex TaqMan qPCR assay amplifying *MT-ND1* (mitochondrial target) and *B2M* (nuclear target) with a CFX96™ Real-Time PCR Detection System (Bio-Rad). Reactions were performed in triplicates with a total volume of 25 μL, consisting of 12.5 μL iTaq Universal Probes Supermix (Bio-Rad), 0.75 μL *MT-ND1* forward primer (10 μM), 0.75 μL *MT-ND1* reverse primer (10 μM), 0.75 μL *B2M* forward primer (10 μM), 0.75 μL *B2M* reverse primer (10 μM), 0.5 μL *ND1*-HEX probe (10 μM), 0.5 μL *B2M*-FAM probe

(10 μM) and 7.5 μL nuclease-free water (Ambion). The cycling conditions consisted of an initial denaturation at 95 °C for 3 min followed by 40 cycles of denaturation at 95 °C for 10 s and 62.5 °C annealing and extension for 1 min. Probe and primer sequences are in Supplementary table 2. Relative mtDNA copy number was calculated by $\Delta Ct = Ct\, B2M - Ct\, ND1$ per sample. Sample mtDNA copy number calculated relative to a control $\Delta Ct$[53].

## RNA-Seq

**Whole transcriptome sequencing.** RNA was isolated from patient muscle samples or fibroblasts and controls using Trizol (Thermo-Fisher), according to the manufacturer's instructions as described previously[54], and was further treated to remove traces of remaining DNA, using Turbo DNA-free (Ambion). As rRNA is extremely abundant, RNA samples were treated with NEBNext rRNA Depletion kit. However, to be able to analyze rRNA as well, a second library of fibroblast samples was generated without rRNA depletion. RNA-Seq libraries were created using NEBNext Ultra II Directional RNA Library Prep Kit for Illumina, following the manufacturer's protocol and library quality was evaluated using Tapestation. Paired-end sequencing was performed using a read length of up to 50 bp on an Illumina NovaSeq instrument. For genome-wide RNA-Seq analysis, paired end sequenced reads were aligned to the GRCh38 transcriptome using Salmon[55], followed by differential expression analysis by DESeq2[56]. Volcano plots were generated with EnhancedVolcano[57] and GO.db[58] was used for Gene Ontology pathway analysis. For analysis of mitochondrial RNA, the reads were mapped with HiSat2[59] and alignments were filtered to retain only properly mapped pairs, reads were split by template strand, and converted to full-length RNA fragment BED files, before coverage was calculated with BEDtools[60].

## Short read sequencing

RNA was isolated from cultured fibroblasts obtained from a skin biopsy of P4. Sequencing was performed at the Victorian Clinical Genetics Service (Melbourne, Australia). Libraries were prepared using a TruSeq Stranded mRNA Library Prep Kit (Cat. No. RS-122-2101, RS-122-2102, and RS-122-2103). Paired-end sequencing was performed using a read length of up to 150 bp on an Illumina HiSeq 2000 instrument to a achieve a minimum sequencing depth of 100 million paired-end reads. RNA sequencing reads were processed using a Bpipe (Version 0.9.9.6, release 21/07/2018)[61] pipeline for quality control checks, trimming, and alignment. FastQC and Trimmomatic[62] were used for sequencing quality checks and trimming of poor-quality reads. Alignment was performed using STAR (version 2.7.3a, release 08/10/2019,)[63], in two-pass mode for read alignment to the Human Reference Genome Build 38 (excluding "ALT" contigs). Duplicate reads were marked with Picard MarkDuplicates and quantification was performed using FeatureCounts from the R Subread package (version 1.34.7, release 03/01/2019)[64]. Differential expression analysis was performed using the DESeq2 package (version 1.25.9, release 31/07/2019) comparing expression of the affected individual to 23 unrelated control samples[56]. Sashimi plots were prepared using ggsashimi[65].

## CSBII analysis

Standard alignment to the reference genome could potentially create a bias in favour of shorter repeat sequences. Therefore, the available whole genome or whole exome sequencing datasets were re-analyzed. Starting from the Fastq data, quality trimming was performed with TrimGalore!. Alignment to hg38 with adaptations (extra Ns in the repeat) was performed with Bowtie2 (−very-sensitive−local−np 0−no-mixed) to avoid a penalty score for (not) mapping to N.

## Western blot and dipstick activity assays

Protein was extracted from cultured fibroblasts and patient muscle and analyzed by western blotting. Samples were first lysed in RIPA

buffer (Sigma-Aldrich) with cOmplet, Mini, EDTA-free Protease Inhibitor Cocktail (Roche). In addition, muscle samples were homogenized using a TissueRuptor II (Qiagen). Twenty micrograms of protein was run per lane on NuPAGE 4–12% Bis-Tris Protein Gels (Invitrogen), transferred to a PVDF membrane and blocked with 5% milk (tris-buffered saline, skimmed milk powder, 0.1% Tween-20) for 1 h at room temperature. Primary antibodies were incubated in 5% milk overnight at 4 °C, washed, and incubated with appropriate HRP secondary antibodies for 1 h at room temperature. Blots were developed with SuperSignal West Dura Extended Duration Substrate (Thermo Scientific). Primary antibodies used are rabbit anti-TEFM (HPA023788), total Abcam OXPHOS human WB antibody cocktail (ab110411), anti-ND6 (V-16)-R (sc-20510-R), anti-TOMM20 (ab186735), anti-VDAC1 (ab14734), anti-NDUFA9 (ab14713), anti SDHA (ab14715), anti-UQCRC1 (ab110252), anti-MTCO1 (ab14705), anti-ATP5A (ab14748), anti-MRPLS18b (16139-1-AP), anti-MRPL12 (14795-1-AP), anti-MRPL3 (HPA043665), anti-MRPS17 (18881-1-AP), anti-MRPS39 (AP1967b), anti-MRPL45 (ab251748), anti-PDE12 (HPA043171), anti-GRSF1 (NBP1-89488) and anti-vinculin (42H89L44). All antibodies were validated by the manufacturer. The oxidative phosphorylation (OXPHOS) complex I and IV activity in fibroblasts was measured using dipstick activity assays (Abcam), using 30 μg of whole-cell lysates following the manufacturers protocol as described[66].

### Preparation, acquisition and analysis of mass spectrometry data

A total of 20 μg of proteins from fibroblasts (TEFM patient as technical duplicate and single replicates from four control individuals) were quantified with Pierce Protein Assay Kit (Thermo Fisher Scientific) in lysis buffer prior to reduction, alkylation using the PreOmics iST-NHS kit reagents. Protein digestion was performed with trypsin at 1:50 trypsin to protein ratio and according to manufacturer's instructions. Clean peptides from individual samples were individually labelled with TMT10plex TMTs (Thermo Fisher Scientific) at a ratio of 8 μg TMT label to 1 μg protein prior to mixing at a 1:1 ratio. Labelled peptides were cleaned up and eluted per manufacturer's instructions following drying down in a CentriVap Vacuum concentrator (Labconco). Pooled peptides were subjected to high-pH fractionation using the Pierce High pH Reversed-Phase Peptide Fractionation Kit (Thermo Fisher Scientific) into 8 fractions as per manufacturer's instructions. Individual fractions were dried down as previously, and peptides were reconstituted in 2% (v/v) acetonitrile and 0.1% (v/v) TFA for mass spectrometry analysis. Liquid chromatography–coupled tandem mass spectrometry LC-MS/MS was performed on an Orbitrap Lumos mass spectrometer (Thermo Fisher Scientific) with a nanoESI interface in conjunction with an Ultimate 3000 RSLC nanoHPLC (Dionex Ultimate 3000). The liquid chromatography system was equipped with an Acclaim Pepmap nano-trap column (Dionex-C18, 100 Å, 75 μm × 2 cm) and an Acclaim Pepmap RSLC analytical column (Dionex-C18, 100 Å, 75 μm × 50 cm). The tryptic peptides were injected to the trap column at an isocratic flow of 5 μl/min of 2% (v/v) acetonitrile containing 0.1% (v/v) formic acid for 5 min, applied before the trap column was switched in-line with the analytical column. The eluents were 5% DMSO in 0.1% v/v formic acid (solvent A) and 5% DMSO in 100% v/v acetonitrile and 0.1% v/v formic acid (solvent B). The flow gradient was (1) 0–6 min at 2% B; (2) 6–95 min at 2–23% B; (3) 95–105 min at 23–40% B; (4) 105–110 min at 40–80% B; (5) 110–115 min at 80–80% B; and (6) 115–117 min at 80%–2% and equilibrated at 2% B for 10 min before the next sample injection. The mass spectrometer was operated in positive-ionization mode with spray voltage set at 1.9 kV and source temperature at 275 °C. The mass spectrometer was operated in the data-dependent acquisition mode mass spectrometry spectra scanning from $m/z$ 375 to 1500 at 120,000 resolution with an AGC target of 400,000. The "top speed" acquisition method mode (3-s cycle time) on the most intense precursor was used, whereby peptide ions with charge states ≥2–7 were isolated with an isolation window of 0.7 m/z and fragmented with high energy collision mode with stepped collision energy of 35 ± 5%. Fragment ion spectra were acquired in Orbitrap at 50,000 resolution. Dynamic exclusion was activated for 45 s.

Raw files were processed using the MaxQuant platform (version 1.6.10.43)[67] and searched against the reviewed UniProt human database (March 2021) containing canonical and isoform sequences using default settings for a TMT 10plex experiment with the following modifications: deamination (of asparagine and glutamine) was added as variable modification, and a mass shift of +113.084 on cysteine was included as a fixed modification as PreOmics kit recommendation. Correction factors for the TMT kit product: A37725 and lot number: UB278774 were entered into the search parameters. The proteinGroups.txt output from the search was analyzed in Perseus (version 1.6.14.0)[68]. TMT reporter intensity corrected values for TEFM replicates (126C, 127N) and controls (129C, 130N, 130C, 131N) were log$_2$-transformed and entries annotated by MaxQuant as "potential contaminant," "reverse," and "only identified by site" were removed from the dataset (Supplementary Dataset). Categorical annotation of the TMT intensities into groups (TEFM, controls) was performed and valid values were filtered to have a minimum 2 in each group. Two sample t-test was performed between TEFM group against controls group using t-test (p-value 0.05) for truncation. Scatter plot function was used to plot volcano plot with fold change of 1.5 (log$_2$ = 0.585) and p-value 0.05. The relative complex abundance of the OXPHOS complexes and mitoribosome subunits were plotted with an in-house R script which calculated the difference between the TEFM and control patients for each subunit identified by more than one peptide as indicated by gene name. The mean and standard deviation were then calculated, along with the 95% confidence interval based on the t-statistic for each complex (calculated from the difference between the control and patient samples), along with a paired t-test which calculated significance between the control and patient for each complex. TEFM protein was detected via a single peptide and the MS/MS spectrum was plotted using the MaxQuant platform (version 1.6.10.43).

### Recombinant protein expression and purifications

Human WT TEFM (UniProt: Q96QE5, residues 36–360, tagged with a N-terminal 6× His-tag) and all mutant variants, were expressed recombinantly in KRX Escherichia coli grown in Terrific Broth (TB) supplemented with 8 g/l glycerol, by the addition of 0.2% rhamnose and 0.5 mM isopropyl β-D-1-thiogalactopyranoside (IPTG)[13]. Induction was maintained at 16 °C for 18 h. TEFM mutant variants were generated by site-directed mutagenesis of the expression plasmid containing the WT TEFM sequence, codon optimized for E. coli, using the Quick-Change Lightning site-directed mutagenesis kit (Agilent Technologies) according to the manufacturer´s recommendations. Primers used in mutagenesis are listed in Supplementary Table 3. TEFM and mutant derivatives thereof were purified over a HIS-Select Nickel Affinity Gel (Sigma-Aldrich) with an imidazole gradient (10–500 mM) in buffer A (100 mM HEPES pH 8.0, 10% glycerol and 1 mM DTT) supplemented with 0.5 M NaCl. The 6×His-tag was cleaved through incubation with tobacco etch virus (TEV) protease, followed by a Nickel Affinity chromatography. The flow through was collected and applied to a HiTrap Heparin HP column (GE Healthcare), which was developed with a linear NaCl gradient (0.2–1.0 M NaCl in Buffer A). TEFM containing fractions were applied to a HiTrap SP HP column (GE Healthcare) and eluted with a NaCl gradient (0.2–1 M in Buffer A). POLRMT lacking the mitochondrial targeting sequence (residues 1–42) and containing an N-terminal MBP tag, was cloned and expressed in ArticExpress E. coli cells (Agilent Technologies) grown in TB media. POLRMT expression was induced by the addition of 1 mM IPTG at 16 °C and induction maintained for 18 h. Cells were harvested in lysis buffer (0.5 M NaCl, 20 mM HEPES pH 8.0, 10 mM β-mercaptoethanol), and after

centrifugation (20,000 × g, 45 min, 4 °C), the supernatant was loaded onto a HiTrap Heparin HP column (GE Healthcare) equilibrated in buffer B (25 mM Tris–HCl pH 8.0, 0.4 M NaCl, 10% glycerol and 1 mM DTT) and eluted with a linear NaCl gradient (0.2–1 M in Buffer B). Fractions containing POLRMT were pooled, MBP tag removed by incubation with TEV protease and dialyzed against Buffer B. PolRMT was further purified through a HiTrap SP HP column (GE Healthcare) eluted with a NaCl gradient (0.2–1 M in Buffer B)[19]. TFAM and TFB2M were cloned independently on pBacPAK9 vectors (Clontech) tagged with an N-terminal 6×His-tag. Recombinant baculoviruses were used to infect *Spodoptera frugiperda* (Sf9) insect cells (Termo Fisher Scientific). Insect cells were collected in lysis buffer (25 mM Tris-HCl pH 8.0, 20 mM β-mercaptoethanol, 0.8 M NaCl) 48 h after infection. Cell lysates were cleared by centrifugation (20,000 × g, 45 min, 4 °C), and supernatant purified through a HIS-Select Nickel Affinity Gel (Sigma Aldrich) eluted in buffer B supplemented with 250 mM imidazole. 6×His-tags were cleaved through incubation with TEV protease, and proteins applied to a HiTrap Heparin HP column (GE Healthcare) eluted with a NaCl gradient (0.2–1.0 M NaCl in Buffer B). A further purification step was performed by loading the proteins onto a HiTrap SP HP column (GE Healthcare) resolved with a NaCl gradient (0.2–1 M in Buffer B)[7,9,19].

### DNA templates
The human LSP region (position 1-477 in mtDNA) was cloned into a pUC18 vector. The guanine (G)-tract of conserved sequence block II (CSB II) present in this construction had the sequence $G_5AG_7$, and unless otherwise specified, this template was used for in vitro transcription reactions. Run-off transcripts of about 3000 or 400 nt were created by linearization of the template with *HindIII* or *BamHI* respectively. (G)-tract length variants of CSB II were generated by inverse PCR of the pUC18-LSP plasmid using the oligonucleotide pairs shown in Supplementary Tables 3 and 4, and recircularization by ligation of these PCR products using T4 ligase (Thermo Fisher).

### In vitro transcription assays
In vitro transcription reactions were performed in 25 µl reaction volume, in buffer containing 25 mM Tris–HCl pH 8.0, 10 mM $MgCl_2$, 64 mM NaCl, 100 µg/ml BSA, 1 mM DTT, 400 µM ATP, 150 µM GTP, 150 µM CTP, 10 µM UTP, 0.02 µM α–$^{32}$P UTP (3000 Ci/mmol), and 4 U RNase inhibitor Murine (New England Biolabs). Each reaction also contained 4 nM of DNA template, 20 nM POLRMT, 30 nM TFB2M, and 120 nM TFAM. The concentrations of TEFM variants (calculated as a dimer) used are indicated in the figure legends. All reactions were set up on ice, initiated by the addition of ribonucleotides and incubated at 32 °C for 5 min, unless otherwise stated. Reactions were stopped by the addition of stop buffer (10 mM Tris–HCl pH 8, 200 mM NaCl, 1 mM EDTA, and 100 µg/ml proteinase K) followed by incubation at 42 °C for 45 min. Transcripts were purified by ethanol precipitation and pellets resuspended in loading buffer (98% formamide, 10 mM EDTA, 0.025 % xylene cyanol, and 0.025 bromophenol blue). Time-course experiments were performed in a total reaction volume of 125 µl per condition and at the indicated times, 25 µl were removed and the reaction terminated with stop buffer. Reaction products were separated on a denaturing 4% polyacrylamide gel followed by exposure on Phosphorimager and quantified using the programme Multi Gauge.

### Zebrafish experiments
**Zebrafish morpholino injections.** The UCSC database[69] (http://genome.ucsc.edu/) revealed one orthologue of *TEFM* in zebrafish; *tefm* (GRCz11/danRer11 assembly). After confirming expression of *tefm* throughout the first 5 days post fertilization (dpf), an antisense morpholino oligonucleotide (MO) targeting the splice acceptor site of intron 1/exon 2 *tefm* (5′–3′ ATGTCCTGCAATCAATCAAACATTT,

NM_001080026.1) was designed and synthesized by Gene Tools LLC (USA) and a control MO with fluorescein tag (targets a human beta-globin mutation, 5′–3′ CCTCTTACCTCAGTTACAATTTATA). MOs were diluted to 2 ng/nl in Danieau buffer (58 mM NaCl, 5 mM HEPES, 0.7 mM KCl, 0.6 mM $Ca(NO_3)_2$, 0.4 mM $MgSO_4$; pH 7.6) and supplemented with phenol red. We injected 4 ng of MO into the yolk sac of embryos at the 1-cell stage. Embryos were maintained at 28.5 °C in blue water (system water with 0.1 µg/ml Methylene Blue) for up to 2 dpf and survival recorded daily. At 2 dpf zebrafish were assessed imaged using a Leica EZ4 W stereomicroscope and head angle and length measured using Fiji (ImageJ)[70].

### Chorion movement analysis in zebrafish
At 1 dpf (24 h post fertilization), zebrafish were recorded in their chorions for 1 min at 30 frames/s using a Leica EZ4 W stereomicroscope. Videos were analyzed using DanioScope software (Noldus Information Technology Inc., Leesburg, VA) to automatically assess burst activity, duration of bursts and burst count/minute.

### RNA isolation, cDNA synthesis, RT-PCR in MO zebrafish
RNA was isolated from pools of around 20, 2 dpf zebrafish (control MO and *tefm* MO-injected) following removal of chorions with pronase (Streptomyces griseus, Roche,1 mg/ml in blue water). Zebrafish were washed 3 times with blue water, euthanized with a 1:1 ratio of fresh system water:4 mg/ml tricaine methanesulfonate (MS-222) (Sigma) then frozen at −80° for 1 week. Fish were homogenized in RLT buffer (RNeasy mini kit, Qiagen) using 5 mm stainless steel beads with a TissueLyser II (Qiagen) at 25 Hz for 2 min. RNA was then isolated following the RNeasy kit manufacturer's instructions, including on-column DNase digestion. RNA was measured using a Nanodrop ND-1000 and 1 µg used for cDNA synthesis according to manufacturer's instructions (5X All-In-One RT MasterMix, abm). Reverse-transcriptase PCR (RT-PCR) was performed to check for *tefm* gene expression and knockdown success in MO-treated embryos, using MyTaq™ DNA Polymerase (Meridian Bioscience) and primers listed in Supplementary Table 5.

### Protein isolation from zebrafish
Zebrafish were euthanized at 2 dpf with a 1:1 ratio of fresh system water:4 mg/ml tricaine methanesulfonate (Sigma), then frozen at −80° for 1 week in pools of around 20 fish per condition, prior to protein isolation. Embryos were deyolked in 1/2 Ginzburg Fish Ringers without Calcium (55 mM NaCl, 1.8 mM KCl, 1.25 mM $NaHCO_3$) by shaking for 5 min at 1100 rpm (Thermomixer, Eppendorf). Samples were pelleted by centrifugation at 300 x g for 3 min and resulting pellet resuspended in lysis buffer (RIPA buffer with protease inhibitor tablet (Roche)). After 30 min constant agitation, samples were centrifuged for 30 min at 13,000 × g, both steps performed at 4 degrees. Supernatants were collected and protein levels for control and *tefm*-MO zebrafish quantified using a DC protein assay (Bio-Rad) according to manufacturer's instructions.

### Western blotting in zebrafish
Protein isolation is described in the supplementary methods. Samples were prepared with 4× Laemmli Sample Buffer (Bio-Rad) and 30 µg protein loaded onto 12% hand-cast tris-glycine gels, transferred to a mini PVDF membrane with a Trans-Blot Turbo transfer system (Bio-Rad) using the standard 30 min protocol. Membranes were probed with antibodies to recognize components of the OXPHOS system: NDUFA9 (1:500, ab14713, Abcam), MTCO1 (1:500, ab14705, Abcam), porin/VDAC1 (1:1000, ab15895, Abcam), ATP5A (1:1000, ab14748, Abcam), and β-actin (1:2000, A1978, Sigma-Aldrich). Membranes were incubated with fluorescent secondary antibodies: IRDye 680RD Goat Anti-Rabbit and IRDye 800RD Goat Anti-Mouse (1:5000, 926-68071 and 926-32210 respectively, LI-COR Biosciences). Blots were imaged

using a LI-COR Odyssey CLx imaging system and analyzed with LI-COR image studio.

## Immunofluorescent staining, imaging and analysis of zebrafish neuromuscular junctions

Whole mount staining of 2 dpf zebrafish NMJs was performed as described by O'Connor et al.[31] Two dpf fish were first dechorinated before being euthanized with tricaine methanesulfonate. Fish were then fixed overnight in 4% paraformaldehyde in phosphate buffered saline (PBS) at 4 °C. To visualize motor neurons, a mouse anti-synaptic vesicle protein 2 (SV2) antibody (1:200, AB_2315387, Developmental Studies Hybridoma Bank) was applied overnight at 4 °C on a shaker. Following washing in PBS with 0.1% tween, a secondary antibody (Alexa Fluor 594 IgG goat anti-mouse, 1:500, Life Technologies, Waltham, MA, USA) was applied for detection. Following this, an Alexa Fluor 488-α-bungarotoxin conjugate (1:1000, B13422, Invitrogen) was applied for 2 h at room temperature to visualize acetylcholine receptors (AChRs). All antibodies were diluted in 5% horse serum in PBS with 0.1% tween. WT un-injected fish were used for comparison, as our control MO has a fluorescein tag. Slides were blinded prior to imaging and unblinded after completion of image analysis. Z-stack images encompassing the depth of the midsection of zebrafish tail were obtained using a 20× air objective on an LSM800 confocal microscope.

Analysis of NMJ structure was performed as described by O'Connor et. al.[31], using Fiji (ImageJ, Madison, WI, USA). The average size of SV2-positive and α-bungarotoxin-positive clusters and the number of clusters per 100 μm² were measured using maximum intensity projections. Thresholding was applied for the relevant channel, images despeckled, then each myotome analyzed independently using the selection tool to delete the rest of the image. The 'analyze particles' function was used to select all clusters in the myotome of interest and this provided a number of clusters and area of each cluster. Number of clusters per 100 μm² was derived from number of clusters per myotome and the area of the myotome. Data was collected from at least 4 myotomes per fish and is presented as average value per fish.

Co-localization analysis between SV2 and αBTx was performed on maximum intensity projections using the 'JACoP' Fiji plugin[71]. Briefly, individual myosepta or myotomes were outlined for analysis and then each fluorophore was subject to manual thresholding to remove background. The Mander's correlation coefficient was calculated to give a value between 0 and 1, reflecting the degree of co-occurrence of signals between both SV2 and α-bungarotoxin, and also α-bungarotoxin and SV2.

## Zebrafish CRISPR/Cas9 tefm (F0) knockout generation

Following the previously published method[72], three predesigned CRISPR RNAs (crRNAs) targeting exon 2 and exon 3 of zebrafish *tefm* were chosen from Integrated DNA Technologies, Inc (IDT) (Supplementary Fig. 10a and Supplementary Table 6) Briefly, each crRNA (200 μM) was annealed to universal tracrRNA (200 μM) (IDT) in nuclease free duplex buffer (IDT) to form three individual guide RNAs (gRNA). Once annealed each gRNA was assembled into a Cas9/gRNA ribonucleoprotein (RNP) with 57 μM Alt-R® S.p. Cas9 Nuclease V3 (IDT) to a final RNP concentration of 28.5 μM. Before injection all three RNPs were pooled together to form the final injection mix. 1nL of injection mix was injected into the yolk of TLF wild type zebrafish, no older than 20 min post fertilization from three separate clutches. Uninjected clutch mates were collected for controls. All embryos were subsequently kept at 28 °C in E3 medium (5 mM NaCl, 0.17 mM KCl, 0.33 mM CaCl₂, 0.33 mM MgSO₄) until 5 dpf. Embryos were monitored for defects and dead embryos removed promptly. At 5 dpf, zebrafish were humanely culled and pools of 10 larvae were collected and frozen at −80 °C.

## Confirmation of CRISPR/Cas9 tefm (F0) knockout generation

DNA was extracted from both 5 dpf, injected and uninjected (control) larvae pools (10 per pool), using a DNeasy blood and tissue kit (Qiagen). PCR reactions were performed using primers flanking the three different mutagenesis target regions (primer sequences in Supplementary Table 6). To establish if mutagenesis of the target regions was successful a heteroduplex mobility assay (HMA) was performed as described by Sorlien et. al.[73] PCR products were first purified with a QIAquick PCR Purification kit (Qiagen) before heating to 100 °C for 3 min to denature the DNA. Samples were then left to cool to room temperature for 1 h to allow the formation of heteroduplex products. Samples were subsequently loaded into Novex™ TBE Gels, 20% (Thermo Fisher Scientific) and run at 150 V for 3 h. Gels were post-stained in 3X GelRed Nucleic Acid Gel Stain (Biotium) solution for 1 h at room temperature before being imaged on an Amersham Imager 680 (Cytiva). Successful mutagenesis was confirmed by the presence of multiple high weight bands not present in the uninjected controls.

## RNA isolation, cDNA synthesis, RT-qPCR in CRISPR/Cas9 tefm (F0) knockout zebrafish

RNA was extracted from pools of 10, 5 dpf zebrafish (injected or uninjected) that had been culled using MS-222 and frozen at −80 °C. Samples were first homogenized using polypropylene pestles in TRIzol Reagent (Invitrogen). Followed by extraction according to the manufacturers protocol. Contaminating genomic DNA was removed from RNA samples with a TURBO DNA-*free* kit (Invitrogen) following the rigorous DNase treatment protocol. RNA concentration was measured using a Nanodrop-2000 and cDNA synthesized using 500 ng of RNA according to the manufacturers protocol (High-Capacity cDNA Reverse Transcription Kit, Applied Biosystems). RT-qPCR reactions were performed with a Bio-Rad CFX96 system in triplicates using the following mix per reaction: 1 μL cDNA (1/10 dilution), 9 μL nuclease free water (Ambion), 1.25 μL forward primer (10 μM), 1.25 μL reverse primer (10 μM) and 12.5 μL SsoAdvanced Universal SYBR Green Supermix (Bio-Rad). RT-qPCR programme: initial denaturation of 95 °C for 3 min followed by 39 cycles of 95 °C (10 s) and 58 °C (1 min). Following this, melt curve analysis was performed, ramping from 65 °C to 95 °C in 0.5 °C increments for 5 s. Three biological replicates (pools of 10, 5 dpf zebrafish) were used per experimental group. Fold change was calculated via the $2^{-\Delta\Delta Ct}$ method relative to *eelf1a1la*. Primer sequences are found in Supplementary Table 6.

## Spontaneous movement analysis of 5dpf tefm − (F0) knockout zebrafish

At 5dpf, 48 uninjected control larvae and 48 *tefm* − (F0) knockout larvae were placed into individual wells of a 96 well transparent microtiter plate in 200 μL of E3 medium. Using a Zantiks MWP (Zantiks. Ltd) system, the distance moved (mm) of each larva was recorded over the period of 80 min while being exposed to either light or darkness for alternating periods of 10 min. The movement during the first 5 min of light and dark cycle 2–4 was used in the final movement analysis to allow for initial habituation to the protocol and to avoid stimulus desensitization.

**Zebrafish experiments.** Statistical analysis was performed using GraphPad Prism software (v9, BD Biosciences, San Jose, CA, USA). Outliers were removed using the ROUT method (set at 1%), then data tested for normal distribution and, from these results, either a non-parametric Mann-Whitney test or parametric test unpaired *t*-test were applied. For colocalization analysis where data for 6 separate myotomes/myosepta were collected for each fish, nested *t*-tests were performed. Statistical significance was taken as $p < 0.05$ and n numbers are listed in the results section.

**Statistics and reproducibility.** Unless stated otherwise, whenever experimentally possible and reasonable, at least three independent replicates were performed.

## Reporting summary

Further information on research design is available in the Nature Portfolio Reporting Summary linked to this article.

## Data availability

The authors declare that the data supporting the findings of this study are available within the paper and its supplementary information files. Accession number for TEFM cDNA used in the study is NM_024683.4. The structure of TEFM (PDB ID: 5OLA) was sourced from PDB (https://www.rcsb.org). The RNASeq fastq files can be accessed on GEO (GSE185245). Diagnostic next generation sequencing data can be made available by the authors on request as this is case sensitive. The mass spectrometry proteomics data has been deposited in the ProteomeXchange Consortium via the PRIDE partner repository under the accession code PXD030427. Project Webpage: FTP Download: ftp://ftp.pride.ebi.ac.uk/pride/data/archive/2022/11/PXD030427 Source data are provided with this paper. All uncropped blots are supplied in the source data file or in the supplementary data file. All data is available with this submission, apart from proprietary scripts.

Reference without DOI:

R Core Team (2020). R: A language and environment for statistical computing. R Foundation for Statistical Computing, Vienna, Austria. URL https://www.R-project.org/ Blighe K., Rana S., Lewis M. EnhancedVolcano: publication-ready volcano plots with enhanced colouring and labelling. 2019. https://github.com/kevinblighe/EnhancedVolcano

Carlson M (2019). GO.db: A set of annotation maps describing the entire Gene Ontology.

Wickham H (2016). ggplot2: Elegant Graphics for Data Analysis. Springer-Verlag New York. ISBN 978-3-319-24277-4, https://ggplot2.tidyverse.org. Source data are provided with this paper.

## Code availability

For analysis of the WES data in Patient 5 we used a home-made pipeline built on the following code: https://github.com/frankMusacchia/VarGenius.

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

## Acknowledgements

L.V.H, P.R-G., C.A.P and M.M. were supported by Medical Research Council, UK (MC_UU_00015/4). H.L. receives support from the Canadian Institutes of Health Research (Foundation Grant FDN-167281), the Canadian Institutes of Health Research and Muscular Dystrophy Canada (Network Catalyst Grant for NMD4C), the Canada Foundation for Innovation (CFI-JELF 38412), and the Canada Research Chairs programme (Canada Research Chair in Neuromuscular Genomics and Health, 950-232279). E.O. is supported by an AFM-Téléthon postdoctoral fellowship. CMG receives support from the Swedish Research Council (2017-01257), the Swedish Cancer Foundation (2017-631), the Knut and Alice Wallenberg Foundation, and grants from the Swedish state under the agreement between the Swedish government and the county councils, the ALF agreement (ALFGBG-728151). R.H. is a Wellcome Trust Investigator (109915/Z/15/Z), who receives support from the Medical Research Council (UK) (MR/N025431/1 and MR/V009346/1), the European Research Council (309548), the Newton Fund (UK/Turkey, MR/N027302/1), the Addenbrookes Charitable Trust (G100142), Evelyn Trust, Lily Foundation, Stoneygate Fund and an MRC strategic award to establish an International Centre for Genomic Medicine in Neuromuscular Diseases (ICGNMD) MR/S005021/1. This research was supported by the NIHR Cambridge Biomedical Research Centre (BRC-1215-20014). The views expressed are those of the authors and not necessarily those of the NIHR or the Department of Health and Social Care. The study was further supported by the Horizon 2020 research and innovation programme via grant 779257 "Solve-RD". Data were analysed using the RD-Connect Genome-Phenome Analysis Platform developed under FP7/2007-2013 funded project (grant agreement no 305444) and funding from European Joint Programme in Rare Disease (EJP-RD) and INB/ELIXIR-ES. This research was also supported by a New South Wales Office of Health and Medical Research Council Sydney Genomics Collaborative grant (J.C.), NHMRC project grants (1026891 to J.C., 1140906 to D.A.S. and 1164479 to D.R.T, J.C. and D.A.S.), and NHMRC research fellowships (1102896 to D.R.T. and 1140851 to D.A.S.). We are grateful to the Crane and Perkins families for their generous financial support. The research conducted at the Murdoch Children's Research Institute was supported by the Victorian Government's Operational Infrastructure Support Program. We thank the Kinghorn Centre for Clinical Genomics for assistance with production and processing of genome sequencing data. The Chair in Genomic Medicine awarded to J.C. is generously supported by The Royal Children's Hospital Foundation. D.H.H. is supported by the Melbourne Research Scholarship and the Mito Foundation top-up Scholarship. G.H. is supported by the Ochsner MD-PhD Scholarship. We thank the Bio21 Mass Spectrometry and Proteomics Facility (MMSPF) for the provision of instrumentation, training, and technical support, and the Mito Foundation for the provision of instrumentation through the large equipment grant support scheme. This study was supported by Telethon Foundation, Telethon Undiagnosed Diseases Program (TUDP, GSP15001). This study was in part generated within the European Reference Network ITHACA and EURO-NMD. MedGenome Labs Ltd, Bangalore is acknowledged for Sanger sequencing of parental samples.

## Author contributions

Conceptualization: L.V.H., H.L., R.H., M.M. Formal analysis: R.R., J.C., G.H., C.S., D.H.H., D.A.S., R.H. M.M. Funding acquisition: J.C., D.R.T., D.A.S., H.L., R.H., M.M. Investigation: L.V.H., E.O., H.D., B.M., K.P., G.A., A.A.F., M.B., M.B., D.B., N.B.P., G.C., N.J.C., N.D,. H.G., G.H., H.H., G.L., K.M., D.M., B.N., C.O., C.A.P., V.P.K., V.P., R.R., P.R.G., C.S., S.V., M.S.Z., A.Z., D.R.T., D.A.S., R.M., J.C., C.G., A.N., H.L., M.M., R.H. Methodology: R.R., G.H., D.H.H., N.J.C., E.O., P.R.G., B.M. Project administration: R.H., M.M. Resources: J.C., C.S., D.A.S. Supervision: J.C., D.R.T., D.A.S., H.L., R.H., M.M. Validation: R.H., M.M. Visualization: R.R., G.H., D.H.H., N.J.C., PRG Writing—original draft: L.V.H., E.O., H.L., R.H., M.M. Writing—review & editing: All authors.

## Competing interests

The authors declare the following competing interests: M.M. is a founder, shareholder and member of the Scientific Advisory Board of Pretzel Therapeutics, Inc., L.V.H. is founder and director of NextGenSeek Ltd. The remaining authors declare no conflict of interest.

## Additional information

**Lindsey Van Haute** [1,19], **Emily O'Connor** [2,3,19], **Héctor Díaz-Maldonado** [4,19], **Benjamin Munro** [5,19], **Kiran Polavarapu** [2,3,6], **Daniella H. Hock** [7], **Gautham Arunachal** [8], **Alkyoni Athanasiou-Fragkouli** [9], **Mainak Bardhan** [6], **Magalie Barth** [10], **Dominique Bonneau** [10], **Nicola Brunetti-Pierri** [11], **Gerarda Cappuccio** [11], **Nikeisha J. Caruana** [7,12], **Natalia Dominik** [9], **Himanshu Goel** [13], **Guy Helman** [14], **Henry Houlden** [9], **Guy Lenaers** [10], **Karine Mention** [15], **David Murphy** [9], **Bevinahalli Nandeesh** [6,16], **Catarina Olimpio** [5], **Christopher A. Powell** [1], **Veeramani Preethish-Kumar** [6], **Vincent Procaccio** [10], **Rocio Rius** [14,17], **Pedro Rebelo-Guiomar** [1], **Cas Simons** [14], **Seena Vengalil** [6], **Maha S. Zaki** [18], **Alban Ziegler** [10],

**David R. Thorburn** [14,17], **David A. Stroud** [7,14], **Reza Maroofian**[9], **John Christodoulou** [14,17], **Claes Gustafsson** [4], **Atchayaram Nalini**[6], **Hanns Lochmüller**[2,3,20], **Michal Minczuk** [1,20] ✉ **& Rita Horvath** [5,20] ✉

[1]MRC Mitochondrial Biology Unit, University of Cambridge, Cambridge CB2 0XY, UK. [2]Children's Hospital of Eastern Ontario Research Institute, University of Ottawa, Ottawa, ON, Canada. [3]Division of Neurology, Department of Medicine, The Ottawa Hospital, Ottawa, ON, Canada. [4]Department of Biochemistry and Cell Biology, University of Gothenburg, SE-405 30 Gothenburg, Sweden. [5]Department of Clinical Neurosciences, School of Clinical Medicine, University of Cambridge, Cambridge, UK. [6]Department of Neurology, National Institute of Mental Health and Neurosciences, Bengaluru, India. [7]Department of Biochemistry and Pharmacology, Bio21 Molecular Science and Biotechnology Institute, University of Melbourne, 30 Flemington Road, Parkville, VIC 3052, Australia. [8]Department of Human genetics, National Institute of Mental Health and Neurosciences, Bengaluru, India. [9]UCL London, Department of Neuromuscular Disorders, Institute of Neurology, University College London, London, UK. [10]Department of Genetics, Mitovasc INSERM 1083, CNRS 6015, University Hospital of Angers, Angers, France. [11]Department of Translational Medicine, University of Naples Federico II, Via s. Pansini, 5, 80131 Naples, Italy. [12]Institute for Health and Sport (IHES), Victoria University, Melbourne, VIC 3011, Australia. [13]Hunter Genetics, Waratah, University of Newcastle, Callaghan, NSW 2298, Australia. [14]Murdoch Children's Research Institute, 50 Flemington Road, Parkville, VIC 3052, Australia. [15]Pediatric Inherited Metabolic Disorders, Hôpital Jeanne de Flandre, Lille, France. [16]Department of Neuropathology, National Institute of Mental Health and Neurosciences, Bengaluru, India. [17]Department of Paediatrics, University of Melbourne, Parkville, VIC 3010, Australia. [18]Clinical Genetics Department, Human Genetics and Genome Research Division, National Research Centre, Cairo 12311, Egypt. [19]These authors contributed equally: Lindsey Van Haute, Emily O'Connor, Héctor Díaz-Maldonado, Benjamin Munro. [20]These authors jointly supervised this work: Hanns Lochmüller, Michal Minczuk, Rita Horvath. ✉e-mail: michal.minczuk@mrc-mbu.cam.ac.uk; rh732@medschl.cam.ac.uk

