## [Peer Review File · Nature Communications]

REVIEWER COMMENTS

Reviewer #1 (Remarks to the Author):

The new genetic syndrome described in the paper is well evaluated, with convincing functional studies and modelling. My major concern is the description of the phenotype and the clinical definition of this new mitochondrial disorder group.

7 patients are described; The phenotype is strikingly variable; but the description is very difficult to follow.

- 1) What was the most common feature, and most common metabolic finding?
- 2) How many patients have multisystem involvement and how many only neurological?
- 3) How many patients had elevated lactate levels during the course of disease?
- 4) How many patients had developmental delay and intellectual disability? Was there any regression?
- 5) What is “slowly progressive developmental delay” ; does it mean regression?
- 6) Patients are described multiple times in the same section (“severe, progressive epileptic encephalopathy with myoclonic jerks P4”, later “P1, P2 and P4 had seizures of variable severity”, later “While both siblings had seizures, P2 had additional features of moderate intellectual disability with tremor and ataxia”). Are there 3 patients with seizures, or 4? (“P3 died at 1 month of age without clinical signs of epileptic activity, and P5, P6 and P7 had no clinical or electrophysiological evidence of epilepsy”). Did P3 had abnormal EEG, or why is he mentioned separately?
- 7) “Two Indian brothers (P1, P2) had a childhood-onset fatigable muscle weakness” what is fatigable? Myasthenic or exercise intolerance? “an abnormal repetitive stimulation” is this detected by exam or EMG? Is the finding comparable to classic myasthenia? Is it comparable to congenital myasthenia? Were other drugs of congenital myasthenia/myasthenia trialed?
- 8) What could be the reason of the finding of OXPHOS deficiency in some patients but not in others? Were there any other mitochondrial biomarkers abnormal in the 4 patients having normal OXPHOS measurements?
- 9) What could be the reason for normal mitochondrial phenotype in the animal model, and were there additional mitochondrial functional tests performed?

Minor concerns

-Is also recommend using the description “brothers of Indian ancestry” instead of Indian brothers.

- “family history was compatible with an autosomal recessive inheritance pattern although consanguinity was only reported in one family.

Why do the authors emphasize “although”?

Table: French is not an ethnic group, recommend to use country of origin instead but for deidentification probably Caucasian, Middle eastern etc is more appropriate

Reviewer #2 (Remarks to the Author):

The authors have identified a small cohort of patients with mutations in TEFM and provided compelling data that this impacts mitochondrial transcription elongation. Mutations in TEFM generate a phenotypically diverse disease. Disease modeling in zebrafish generally supports that TEFM impacts mitochondria and NMJs in vivo (although there are some caveats see below). In general, this is an important contribution to the rare disease field.

The zebrafish studies round the manuscript and provide evidence for “genotype (morpholinos are used) ”-phenotype correlation. There are a couple of concerns with the interpretation however. The authors inject a single morpholino against TEFM and show that the transcript is reduced. There is no dose response curve. While the phenotype may be what the authors expect, this is not an acceptable approach to determine gene function. Multiple approaches could be used to bolster their data: Rescue of the phenotype by expression of *tefm*, efficient F0 CRISPR-mediated mutagenesis, injection of a second morpholino with a different target site that generates the same phenotype, rescue of the zebrafish with the drug that improved neuromuscular phenotypes in humans could even be used – but some other data seem necessary.

The NMJ data are compelling but there are a couple of questions. One, many of the morphometric analyses are done by hand (going back to the referenced paper because this is not included in the methods), but there is no mention of blinding of the images prior to manual measurements. More importantly, there is no discussion of the different types of innervation, the effects of the TEFM morpholino on said innervation, and the impact of the results. The myoseptal innervation is slow-twitch

muscle innervation, the distributed innervation is fast-twitch muscle innervation. How do disruptions in these two types of innervation impact the interpretation of TEFM function? Furthermore, the most obviously disrupted innervation is the focal innervation that is reflective of most mammalian NMJs. How does this impact their interpretation?

Reviewer #3 (Remarks to the Author):

In this paper the authors describe the first cohort of patients with mutations in the mitochondrial transcription elongation factor TEFM. In this report they describe the patients phenotypes and genetic analyses, various functional studies using patient muscle biopsies and fibroblasts, in vitro studies on isolated wt and mutant proteins and a TEFM knockdown study using zebrafish. This is an interesting and well executed patient/experimental study that conforms with earlier TEFM functional studies and mouse model.

I have minor and mostly cosmetic comments:

Table 1, row 'intellectual disability' has a comment for the last patient that I assume has been a comment left during the editing phase of the manuscript

It is mentioned that the Lys188Arg mutation shows very little conservation, maybe the authors could mention that in mouse this residue is actually Arg.

At the bottom of pg 9 the following is said: Furthermore, these results also demonstrate that the 288 c.484_489delGAAAGA variant identified in Family 3 (P4), which is also found in Family 4 289 (P6 and P7), leads to reduced activity of mitochondrial complex I.

This sentence was a bit confusing as a last sentence for this section as the part directly above deals with observations in patient skeletal muscle. Perhaps it can be moved up, although it also is somewhat superfluous in my opinion.

I found panels B/F and G/H in Fig 4 initially somewhat confusing. Perhaps in the figure itself the authors can directly indicate which panel concerns fibroblasts and which skeletal muscle

Fig 5, the lines through the bar graphs are very suggestive but what are they based on? Are they trend lines, were they fitted by linear regression, what is the statistics?

In the introduction the involvement for TEFM in relation to the transcription/replication switch at CSBII is clear but the discussion, where it is for example stated 'enhanced

transcription termination at CSBII due to TEFM binding' I initially found very confusing. because it suggests TEFM 'normally' enhances termination, whereas, as I understand it, it normally stimulates transcription to pass through CSBII without termination, as is also clearly shown in Suppl Fig 7.

Last but not least, the CSBII in vitro analysis of course raises the very interesting question of the length variation of the CSBII G-tracts in the various patient families? This data I assume can be extracted quite easily from exome and whole genome sequence data, or alternatively can be determined easily from patient DNA samples by PCR sequencing.

Reviewer #4 (Remarks to the Author):

Mitochondrial dysfunctions are an underappreciated cause of many pathologies. The study by Van Haute and colleagues aims to elucidate molecular causes of neurodegenerative diseases and takes advantage of well-studied mechanistic aspects of mitochondrial transcription. The manuscript is well written and represents an important contribution to the field. The study identifies several mutations in a key transcription regulator, TEFM, and uses multiple approaches to investigate how they affect various aspects of mitochondrial homeostasis. The experiments are performed at a high technical level, and the obtained results advance our knowledge of the causes of neurodegenerative diseases. However, some concerns need to be addressed before this manuscript can be accepted for publication.

1. The conclusions from the in silico analysis of TEFM mutations can be made stronger. First, as the authors correctly suggest in the Discussion, the R34 residue, which is absolutely conserved in mammalian TEFM, is the likely recognition site for the mitochondrial processing peptidase, MPP. Therefore, it is expected that the R34W mutation found in P5 will result in a lack of TEFM processing in mitochondria and a partially unfolded and partially active protein. Second, the discussion of the effects of single mutations and deletions should be refocused. It is unlikely that a single mutation can affect the dimerization of TEFM as the identified mutations are far from the dimer interface formed by the long alpha-helices in the C-terminus of TEFM. Instead, the discussion should focus on the fact that most mutations cluster in an important structural element of TEFM - the linker subdomain. The crystal structure of the polymerase-TEFM complex lacks part of the linker and the NTD, which limits our ability

to predict the effect of every mutation on TEFM function. Therefore, it might suffice to say that the region where the most mutations are found has been previously identified as critical to TEFM function, and mutations there dramatically affect transcription (Hillen et al., Cell 2017) and follow this by the detailed analysis of each mutation.

2. The previous study (Hillen et al., Cell 2017) has identified that mutations in the linker subdomain dramatically affect the anti-termination activity of TEFM but have a limited effect on transcription processivity. Remarkably, the phenotypes of the mutants constructed by the authors also indicate the same trend. It is, therefore, would be logical to emphasize this finding by moving the in vitro anti-termination assay from the Supplement (Fig 7S) into the main figure (Fig 6) of the manuscript. Consistently, one of the processivity assays shown in Fig 6 could be moved into the Supplement.

3. The authors should provide the sequence of the mtDNA CSB2 region for all relevant patients so the outcome of the in vitro anti-termination assays different SCB2 regions (G12 vs. G13 vs. G16, Fig S7) could be discussed in light of the expected anti-termination function of TEFM in vivo. Do patients P1 and P2 have a long G-stretch in the SCB2 region?

4. The most dramatic effect observed in mitochondria of P1 and P2 patients is a 4-5 fold increased copy number of mtDNA (Fig 4H). Yet, the manuscript (lines 297-299) says that "TEFM does not interfere with mtDNA replication". This is a non sequitur. Moreover, considering that the mitochondrial biogenesis is not affected based on experiments in Fig 4, the observed effect cannot simply be attributed to a compensatory mechanism.

5. The observation that mtDNA copy number is dramatically increased in P1 and P2, along with the strong phenotype of the mutants in in vitro anti-termination assays, suggests that transcription termination is affected. Therefore, as it stands, the title of the manuscript does not fully reflect the data and conclusions and should be made more general by eliminating the word "elongation".

6. The authors' data on OXPHOS function in the patients' mitochondria are inconclusive and need clarification.

7. The Western blot of P1 (16 y.o.) and P2 (24 y.o.) fibroblasts indicate a knockout-like level of TEFM. The authors should comment on this in light of a published work showing that TEFM knockouts are embryonically lethal.

8. Related to the above - please provide the source of the anti-TEFM antibodies and their validation using the recombinant TEFM.

RESPONSE TO REVIEWER COMMENTS

Reviewer #1 (Remarks to the Author):

The new genetic syndrome described in the paper is well evaluated, with convincing functional studies and modelling. My major concern is the description of the phenotype and the clinical definition of this new mitochondrial disorder group.

We thank the reviewer for the positive comment. We fully understand the concern about the variability of the phenotype. In fact, this is one interesting characteristic of this study. To make it easier to get a picture about the overall clinical presentation, we added an extra Table (Table 1) summarizing the main features and most common metabolic findings observed in the patients. Table 1 includes the number of patients who presented with multisystem or purely neurological involvement, elevated lactate levels, developmental delay and intellectual disability and epilepsy, etc. We moved the previous very detailed Table 1 to the Supplement, as Supplementary Table 1.

7 patients are described; The phenotype is strikingly variable; but the description is very difficult to follow.

- 1) What was the most common feature, and most common metabolic finding?
- 2) How many patients have multisystem involvement and how many only neurological?
- 3) How many patients had elevated lactate levels during the course of disease?
- 4) How many patients had developmental delay and intellectual disability? Was there any regression?
- 5) What is “slowly progressive developmental delay” ; does it mean regression?
- 6) Patients are described multiple times in the same section (“severe, progressive epileptic encephalopathy with myoclonic jerks P4”, later “P1, P2 and P4 had seizures of variable severity”, later “While both siblings had seizures, P2 had additional features of moderate intellectual disability with tremor and ataxia”). Are there 3 patients with seizures, or 4? (“P3 died at 1 month of age without clinical signs of epileptic activity, and P5, P6 and P7 had no clinical or electrophysiological evidence of epilepsy”). Did P3 had abnormal EEG, or why is he mentioned separately?

The answers to questions 1-6 are now included in the new Table 1.

- 7) “Two Indian brothers (P1, P2) had a childhood-onset fatigable muscle weakness” what is fatigable? Myasthenic or exercise intolerance? “an abnormal repetitive stimulation” is this

detected by exam or EMG? Is the finding comparable to classic myasthenia? Is it comparable to congenital myasthenia? Were other drugs of congenital myasthenia/myasthenia trialed?

In the revised version of the manuscript we have provided a detailed description of these two Indian brothers in the Supplementary material of the paper.

“Patient 1 (P1) and Patient 2 (P2) (Fig. 1A): P1 is a 16-year-old Indian male born to healthy parents, who are not aware of consanguinity. Pregnancy, birth and developmental milestones were normal until age 5 years, when fatigable muscle weakness on the extremities with droopy eyelids and fluctuating restriction of eye movements with occasional double vision were noted. He developed significant disability from 11 years of age with fatigable weakness and difficulty in walking for longer distances without support. Walking for 100 m made him tired and he could not perform any sports. Generalized tonic-clonic seizures occurred (mostly at night) from 12 years of age, requiring combined antiepileptic treatment with Carbamazepine (300 mg/day) and Clobazam (5 mg/day), in the last 6 months he has been seizure-free. P2 is the elder brother of P1, his last examination was at 21 years of age. Similar to his younger brother he presented with fatigable muscle weakness and fluctuating eye movement abnormalities since childhood and also developed nocturnal seizures from 11 years of age. In contrast to P1, he has tremor and intellectual disability (functioning at level of a 5-6 years old) while his muscle weakness and fatigability was stable with little or no progression. Serum lactate was increased in both patients, brain magnetic resonance imaging (MRI) was normal except for an epidermoid cyst in P1, and a focal hyperintense area in left periventricular region in P2 (Fig. 2A). Muscle MRI in patient 1 showed atrophy with fatty replacement of bilateral biceps femoris, semimembranosus & semitendinosus and gastrocnemius muscles (Fig. 2B). In both brothers, skeletal muscle biopsy revealed several cytochrome c oxidase (COX) negative ragged red fibres (RRF) (Fig. 2C) and a severe combined defect of complexes I and IV, but normal respiratory chain (RC) activities in fibroblasts (data not shown). A significant decrement was detected in the younger sibling (Patient 1) on repetitive stimulation who had more severe muscle symptoms, which is a clear indication of a neuromuscular transmission defect (Fig. 2D). Patient 1 was initially started on oral pyridostigmine (up to 180 mg/day) and salbutamol (up to 6 mg/day) in view of the neuromuscular junction (NMJ) defect. He showed significant improvement initially with improved gait (walked 100 m without support) and reduced fatigability. Ptosis and bulbar symptoms also improved, and he was able to do activities of daily living independently or

with minimal support (Supplementary videos 1 and 2). No further improvement was observed on 180 mg/day pyridostigmine and 6 mg/day salbutamol, and he reported headaches, which resulted in compliance issues and he discontinued the treatment. The elder brother never took pyridostigmine or salbutamol as his motor disability is mainly due to occasional gait imbalance and tremors.”

8) What could be the reason of the finding of OXPHOS deficiency in some patients but not in others?

Owing to sample availability, we could only analyze skeletal muscle biopsies of two patients, whereas fibroblasts were available from three additional patients. Skeletal muscle showed severe combined OXPHOS defect and prominent defect of mitochondrial transcription elongation in both muscle samples. In fibroblasts we detected a defect of mitochondrial transcription elongation in all 3 studied cell lines (P2, P3, P6), confirming the pathogenicity of the TEFM variants. There was a mildly significant defect of complex I in one of the three studied patient fibroblasts, while the two other patient fibroblast lines did not show significant changes on immunoblotting of OXPHOS subunits, suggesting that the level of mitochondrial transcription defect has not reached the threshold in fibroblasts to result in reduced enzyme subunit steady state. Tissue specificity is a common finding in mitochondrial disease and the OXPHOS defect often do not present in fibroblasts (Boczonadi et al. 2018; D`Souza et al., 2018). This is in a good agreement with other genetic disease of mitochondrial transcription, similar to the observations reported in patients with POLRMT mutations (Olahova et al., 2021).

Were there any other mitochondrial biomarkers abnormal in the 4 patients having normal OXPHOS measurements?

Additional mitochondrial biomarkers were not tested in these patients.

9) What could be the reason for normal mitochondrial phenotype in the animal model, and were there additional mitochondrial functional tests performed?

Immunoblotting in tefm MO treated fish detected mildly reduced steady-state levels of complex I (non-significant) and significantly reduced steady-state levels of complex IV (Supplementary Figure 9). To address Reviewer’s point and in further support of the

mitochondrial molecular phenotype, in the revised version of the manuscript we show that mitochondrial gene expression is affected in the CRISPR/Cas9 treated zebrafish (Supplementary Figure 10).

Minor concerns

-I also recommend using the description “brothers of Indian ancestry” instead of Indian brothers.

We thank the reviewer for this comment and corrected this wording.

- “family history was compatible with an autosomal recessive inheritance pattern although consanguinity was only reported in one family.

Why do the authors emphasize “although”?

We deleted “although” in this sentence.

Table: French is not an ethnic group, recommend to use country of origin instead but for deidentification probably Caucasian, Middle eastern etc is more appropriate

We thank the reviewer for this comment and corrected this in the Table.

Reviewer #2 (Remarks to the Author):

The authors have identified a small cohort of patients with mutations in *TEFM* and provided compelling data that this impacts mitochondrial transcription elongation. Mutations in *TEFM* generate a phenotypically diverse disease. Disease modeling in zebrafish generally supports that *TEFM* impacts mitochondria and NMJs *in vivo* (although there are some caveats see below). In general, this is an important contribution to the rare disease field.

We thank the reviewer for this very positive assessment of our work.

The zebrafish studies round the manuscript and provide evidence for “genotype (morpholinos are used)”-phenotype correlation. There are a couple of concerns with the interpretation

however. The authors inject a single morpholino against TEFM and show that the transcript is reduced. There is no dose response curve. While the phenotype may be what the authors expect, this is not an acceptable approach to determine gene function. Multiple approaches could be used to bolster their data: Rescue of the phenotype by expression of *tefm*, efficient F0 CRISPR-mediated mutagenesis, injection of a second morpholino with a different target site that generates the same phenotype, rescue of the zebrafish with the drug that improved neuromuscular phenotypes in humans could even be used – but some other data seem necessary.

*We thank the reviewer for pointing out this important issue. We note that we have performed morpholino downregulation of *tefm* in our fish model with two different morpholinos. While one of them was designed to alter splicing, the other one was intended to block the translation of the *tefm* protein. The antibodies available for *tefm* unfortunately do not cross-react with the zebrafish protein hence we were unable to analyze *tefm* steady-state levels upon MO treatment. Therefore, to strengthen the conclusions related to the *tefm* downregulation, we performed CRISPR/Cas9 mutagenesis. To this end, three Cas9/gRNA ribonucoprotein (RNPs) complexes (targeting exon 2 and 3 of zebrafish *tefm*) were co-injected into newly fertilized wild-type zebrafish embryos to generate a *tefm*-knockout in the F0 generation. High resolution melt analysis (HMA) indicated that gRNAs 1 and 2 had effectively induced multiple indels and insertions into exon 2 and 3 of zebrafish *tefm*, indicated by the presence of multiple bands of varying molecular weight in the injected samples (Supplementary Fig. 10B). Mutations in *tefm* of 5 dpf zebrafish impact darkness induced spontaneous movement compared to uninjected controls and this shows some fatigability with time, confirming the phenotype observed in the MO treated fish (pages 13-14).*

The NMJ data are compelling but there are a couple of questions. One, many of the morphometric analyses are done by hand (going back to the referenced paper because this is not included in the methods), but there is no mention of blinding of the images prior to manual measurements.

Images were blinded prior to analysis, this has now been added into the methods section. Analysis is 'done by hand' as we are unaware of an automated way of doing this analysis. However, thresholding, selection of clusters for quantification and co-localization analysis

are performed by ImageJ, only outlining of 'myosepta' or 'myotomes' as regions of interest are performed by hand. This information has now been added to the methods section.

More importantly, there is no discussion of the different types of innervation, the effects of the TEFM morpholino on said innervation, and the impact of the results. The myoseptal innervation is slow-twitch muscle innervation, the distributed innervation is fast-twitch muscle innervation. How do disruptions in these two types of innervation impact the interpretation of TEFM function? Furthermore, the most obviously disrupted innervation is the focal innervation that is reflective of most mammalian NMJs. How does this impact their interpretation?

We thank the reviewer for raising this interesting point. We re-analyzed the co-localization data and split it into fast and slow muscle innervation, however, we see a decrease in co-occurrence of pre and post-synapse components on both fast and slow muscle. This suggests that at least at this gross morphological overview of innervation in the fish there are no muscle type-specific changes due to loss of tefm. We are collaborating with a group to establish electrophysiological analysis of fish larvae NMJs and thus we hope to be able to answer this question more thoroughly in our fish models in the future. However, we feel that that the latter analysis is beyond the scope of this manuscript.

Reviewer #3 (Remarks to the Author):

In this paper the authors describe the first cohort of patients with mutations in the mitochondrial transcription elongation factor TEFM. In this report they describe the patients' phenotypes and genetic analyses, various functional studies using patient muscle biopsies and fibroblasts, in vitro studies on isolated wt and mutant proteins and a TEFM knockdown study using zebrafish. This is an interesting and well executed patient/experimental study that conforms with earlier TEFM functional studies and mouse model.

We thank the reviewer for these very positive comments.

I have minor and mostly cosmetic comments:

Table 1, row 'intellectual disability' has a comment for the last patient that I assume has been a comment left during the editing phase of the manuscript

We thank the reviewer for pointing out this error. We corrected it.

It is mentioned that the Lys188Arg mutation shows very little conservation, maybe the authors could mention that in mouse this residue is actually Arg.

We added this comment to the manuscript on page 7.

At the bottom of pg 9 the following is said: Furthermore, these results also demonstrate that the 288 c.484_489delGAAAGA variant identified in Family 3 (P4), which is also found in Family 4 289 (P6 and P7), leads to reduced activity of mitochondrial complex I.

This sentence was a bit confusing as a last sentence for this section as the part directly above deals with observations in patient skeletal muscle. Perhaps it can be moved up, although it also is somewhat superfluous in my opinion.

We agree with the reviewer and we made the statement on page 9 clearer:

“The second heterozygous variant detected in P4, c.484_489delGAAAGA, which results in the deletion of two amino acids (p.(Glu162_Arg163del)), was also detected in homozygosity in Family 5 (P6 and P7). Furthermore, we demonstrate that the 288 c.484_489delGAAAGA variant leads to reduced activity of mitochondrial complex I in fibroblasts of P4 (Supplementary Fig. 1).”

I found panels B/F and G/H in Fig 4 initially somewhat confusing. Perhaps in the figure itself the authors can directly indicate which panel concerns fibroblasts and which skeletal muscle.

We added “muscle” and “fibroblasts” directly to Figure 4, following the advice of the reviewer.

Fig 5, the lines through the bar graphs are very suggestive but what are they based on? Are they trend lines, were they fitted by linear regression, what is the statistics?

Indeed, the trend lines on Figure 5 were fitted by linear regression. We have updated the figure legend of Fig 5 (and supp Figure S4 and 5) to make this clear.

In the introduction the involvement for TEFM in relation to the transcription/replication switch at CSBII is clear but the discussion, where it is for example stated 'enhanced transcription termination at CSBII due to TEFM binding' I initially found very confusing. because it suggests TEFM 'normally' enhances termination, whereas, as I understand it, it normally stimulates transcription to pass through CSBII without termination, as is also clearly shown in Suppl Fig 7.

We agree with the reviewer and changed the statement in the Discussion on page 17: we deleted “due to TEFM binding” and added “TEFM normally stimulates transcription to pass through CSBII without termination”.

Last but not least, the CSBII in vitro analysis of course raises the very interesting question of the length variation of the CSBII G-tracts in the various patient families? This data I assume can be extracted quite easily from exome and whole genome sequence data, or alternatively can be determined easily from patient DNA samples by PCR sequencing.

We thank the reviewer for this excellent suggestion. As differences in length variation can cause differences in alignment to the reference genome (longer G-tracts might not align), we remapped the reads to a reference genome supplemented with Ns in the G-tracts and mapping settings --np 0, so no penalty score would be given to (not) mapping to an N. The results of this analysis can be found in Supp Table 3 and we have updated the manuscript accordingly. We discuss these finding on page 12.

Reviewer #4 (Remarks to the Author):

Mitochondrial dysfunctions are an underappreciated cause of many pathologies. The study by Van Haute and colleagues aims to elucidate molecular causes of neurodegenerative diseases and takes advantage of well-studied mechanistic aspects of mitochondrial transcription. The manuscript is well written and represents an important contribution to the field. The study identifies several mutations in a key transcription regulator, TEFM, and uses multiple

approaches to investigate how they affect various aspects of mitochondrial homeostasis. The experiments are performed at a high technical level, and the obtained results advance our knowledge of the causes of neurodegenerative diseases.

We thank the reviewer for the positive assessment.

However, some concerns need to be addressed before this manuscript can be accepted for publication.

1. The conclusions from the in silico analysis of TEFM mutations can be made stronger. First, as the authors correctly suggest in the Discussion, the R34 residue, which is absolutely conserved in mammalian TEFM, is the likely recognition site for the mitochondrial processing peptidase, MPP. Therefore, it is expected that the R34W mutation found in P5 will result in a lack of TEFM processing in mitochondria and a partially unfolded and partially active protein.

We thank the reviewer for this comment, and added to the Discussion on page 16:

“The predicted length of the MTS in TEFM is 36 aa, it is possible that Arg34, which is the likely recognition site for the mitochondrial processing peptidase (MPP), is absolutely conserved in mammalian TEFM. MPP recognizes a large variety of MTSs in mitochondrial preproteins and cleaves the single site, often including arginine, at the -2 position^{36,37}. Therefore, it is expected that the p.(Arg34Trp) mutation found in P5 will result in a lack of TEFM processing in mitochondria and a partially unfolded and partially active protein.”

Second, the discussion of the effects of single mutations and deletions should be refocused. It is unlikely that a single mutation can affect the dimerization of TEFM as the identified mutations are far from the dimer interface formed by the long alpha-helices in the C-terminus of TEFM. Instead, the discussion should focus on the fact that most mutations cluster in an important structural element of TEFM - the linker subdomain. The crystal structure of the polymerase-TEFM complex lacks part of the linker and the NTD, which limits our ability to predict the effect of every mutation on TEFM function. Therefore, it might suffice to say that the region where the most mutations are found has been previously identified as critical to TEFM function, and mutations there dramatically affect transcription (Hillen et al., Cell 2017) and follow this by the detailed analysis of each mutation.

We thank the reviewer for this constructive comment and added to the Discussion (page 16):

“It is unlikely that a single mutation can affect the dimerization of TEFM as the identified mutations are far from the dimer interface formed by the long alpha-helices in the C-terminus of TEFM. Most mutations cluster in an important structural element of TEFM - the linker subdomain, which has been previously identified as critical to TEFM function, and mutations there dramatically affect transcription (Hillen et al., Cell 2017).”

2. The previous study (Hillen et al., Cell 2017) has identified that mutations in the linker subdomain dramatically affect the anti-termination activity of TEFM but have a limited effect on transcription processivity. Remarkably, the phenotypes of the mutants constructed by the authors also indicate the same trend. It is, therefore, would be logical to emphasize this finding by moving the in vitro anti-termination assay from the Supplement (Fig 7S) into the main figure (Fig 6) of the manuscript. Consistently, one of the processivity assays shown in Fig 6 could be moved into the Supplement.

We agree with the reviewer and we substituted Figure S7 with the main Figure 6, and the processivity assays from Figure 6 were moved to the Supplement (Supplementary Fig 7). We also emphasized the consistency between Hillen et al., (2017) and our findings (page 12).

3. The authors should provide the sequence of the mtDNA CSB2 region for all relevant patients so the outcome of the in vitro anti-termination assays different SCB2 regions (G12 vs. G13 vs. G16, Fig S7) could be discussed in light of the expected anti-termination function of TEFM in vivo. Do patients P1 and P2 have a long G-stretch in the SCB2 region?

We have added the mtDNA sequences of the CSB2 region for all patients for which the data was available (Supp Table 3) Please see the reply to Reviewer 3.

4. The most dramatic effect observed in mitochondria of P1 and P2 patients is a 4-5 fold increased copy number of mtDNA (Fig 4H). Yet, the manuscript (lines 297-299) says that "TEFM does not interfere with mtDNA replication". This is a non sequitur. Moreover, considering that the mitochondrial biogenesis is not affected based on experiments in Fig 4, the observed effect cannot simply be attributed to a compensatory mechanism.

To address the Reviewer’s comment, we modified the text in page 12: “Given that the mitochondrial mass is not increased in patient cells (Fig. 4), the observed increase in mtDNA

content may not result from the compensatory response due to perturbed mitochondrial gene expression, as observed previously²³. In this view, these results indicate that mutations in TEFM may interfere with mtDNA transcription termination at CSB II leading to enhanced primer formation.”

5. The observation that mtDNA copy number is dramatically increased in P1 and P2, along with the strong phenotype of the mutants in vitro anti-termination assays, suggests that transcription termination is affected. Therefore, as it stands, the title of the manuscript does not fully reflect the data and conclusions and should be made more general by eliminating the word "elongation".

We eliminated the word “elongation” in the title.

6. The authors' data on OXPHOS function in the patients' mitochondria are inconclusive and need clarification.

We could analyze skeletal muscle biopsy of two patients, while fibroblasts were available from one of these patients and two additional patients. Skeletal muscle showed severe combined OXPHOS defect and prominent defect of mitochondrial transcription elongation in both muscle samples. In fibroblasts we could detect a defect of mitochondrial transcription elongation in all 3 studied cell lines (P2, P3, P6), confirming the pathogenicity of the TEFM variants. There was a mildly significant defect of complex I in one of the three studied patient fibroblasts, while the two other patient fibroblast lines did not show significant changes on immunoblotting of OXPHOS subunits, suggesting that the level of mitochondrial transcription defect has not reached the threshold in fibroblasts to result in reduced enzyme subunit steady state. Tissue specificity is a common finding in mitochondrial disease and the OXPHOS defect often do not present in fibroblasts (Boczonadi et al. 2018; D`Souza et al., 2018). This is in line with the expectations in a disease of mitochondrial transcription, similar to seen in patients with POLRMT mutations (Olahova et al., 2021). (see also in response to Reviewer 1)

7. The Western blot of P1 (16 y.o.) and P2 (24 y.o.) fibroblasts indicate a knockout-like level of TEFM. The authors should comment on this in light of a published work showing that TEFM knockouts are embryonically lethal.

We have replaced the western blot figure with a longer exposure of the same membrane, showing a faint band in the patients' fibroblast samples.

We added to the Discussion on page 14:

*“Intercrossing of heterozygous mice produced no viable homozygous knockout (*Tefm* $-/-$) mice, demonstrating that loss of *Tefm* results in embryonic lethality (Jiang et al., 2019). However, very low residual TEFM protein level in P1 and P2 was compatible with life.”*

8. Related to the above - please provide the source of the anti-TEFM antibodies and their validation using the recombinant TEFM.

We thank the reviewer for spotting that this antibody was missing in the materials section. We have added this to the manuscript.

REVIEWER COMMENTS

Reviewer #1 (Remarks to the Author):

The manuscript improved significantly and adding a table to summarize the clinical features was definitely helpful.

I still didn't receive clear information on the phenotype called Fatigable weakness.

Is this exercise intolerance or more myasthenia like symptoms?

Using Pyridostigmine would suggest the later?

However there is no clear statement in the answer or in the supplements. Also I assume an EMG was performed?

Defining the phenotype is really important, and all details are essential in this highly variable presentation.

Reviewer #2 (Remarks to the Author):

The authors diligently addressed reviewer comments.

Reviewer #3 (Remarks to the Author):

The authors have responded appropriately to the concerns of the reviewers. However, some of the changes are very sloppy. In addition, muscle/fibroblasts indications for Figure 4 have not been included, as commented in their response to reviewer 3.

Sloppy changes:

Pg 10 lines 302-304

'In this view, these results indicate that mutations in TEFM may interfere with mtDNA transcription termination at CSB II leading to enhanced primer formation.'

>This again is confusing: I assume the authors mean to say that the TEFM mutations may result in more frequent termination at CSBII and not that they may interfere (suggesting termination is reduced).

Pg 12 373-380

'Given that, the composition of CSB II could have a confounding effect, we analysed the CSBII sequences in the patients and some of their family members (Supplementary Table 3). However, we did not find any substantial

differences in the 5" part of CSB II sequences, suggesting that the sequence of the CSB II G track does not influence the molecular outcomes in the analyzed patients. Consistent with the previously published data 8, we found that mutant TEFM-variants had a reduced ability to prevent premature termination at CSB II, particularly with templates containing longer (G)-tract sequences (Fig. 6C-E).

>Here the order is not very logical considering the part preceding this text block. It would be much more logical to start with the last sentence, 'Consistent with...etc' and then continue with 'Given that the composition.....etc' (komma removed).

Pg17 553-555

The predicted length of the MTS in TEFM is 36 aa, it is possible that Arg34, which is the likely recognition site for the mitochondrial processing peptidase (MPP), is absolutely conserved in mammalian TEFM.

>This sentence doesn't flow and is non-sensical

Reviewer #4 (Remarks to the Author):

The authors provided a thorough response to the criticism and improved the manuscript.

However, an issue raised during the first round of reviews still needs to be addressed to strengthen the importance of the conclusions on TEFM detection in patients' tissues. Since there is no published information on anti-TEFM antibody specificity in Western Blot assays, the authors were requested to provide an experiment demonstrating the detection of recombinant TEFM and TEFM in any somatic cells (or mitochondrial extracts of these cells) with the antibodies used in experiments with patients' tissues. This figure could be presented just in the response to the reviewers' comments.

RESPONSE TO REVIEWER COMMENTS

Reviewer #1 (Remarks to the Author):

The manuscript improved significantly and adding a table to summarize the clinical features was definitely helpful.

I still didn't receive clear information on the phenotype called Fatigable weakness.

Is this exercise intolerance or more myasthenia like symptoms?

Using Pyridostigmine would suggest the later?

However there is no clear statement in the answer or in the supplements. Also I assume an EMG was performed?

Defining the phenotype is really important, and all details are essential in this highly variable presentation.

We modified our description of the muscle weakness and electrophysiology which is shown on Fig. 2B:

“Two brothers of Indian ancestry (P1, P2) had a childhood-onset fluctuating muscle weakness and fatigability, and were clinically diagnosed as possible congenital myasthenic syndrome. The muscle weakness was more progressive in P1 who had myopathic pattern on EMG with fatty replacement of proximal and distal leg muscles (**Fig. 2B**) and an abnormal repetitive stimulation (**Fig. 2C**), suggesting a defect of neuromuscular transmission.” (page 6)

Reviewer #2 (Remarks to the Author):

The authors diligently addressed reviewer comments.

We thank the reviewer for his constructive comments in the first review.

Reviewer #3 (Remarks to the Author):

The authors have responded appropriately to the concerns of the reviewers. However, some of the changes are very sloppy. In addition, muscle/fibroblasts indications for Figure 4 have not been included, as commented in their response to reviewer 3.

We included Figure 4 in this revision and apologise for the mistake in the previous revision.

Sloppy changes:

Pg 10 lines 302-304

'In this view, these results indicate that mutations in TEFM may interfere with mtDNA transcription termination at CSB II leading to enhanced primer formation.'

>This again is confusing: I assume the authors mean to say that the TEFM mutations may result in more frequent termination at CSBII and not that they may interfere (suggesting termination is reduced).

We corrected this sentence on page 10:

“In this view, these results indicate that mutations in TEFM may result in more frequent termination at CSBII.”

Pg 12 373-380

'Given that, the composition of CSB II could have a confounding effect, we analysed the CSBII sequences in the patients and some of their family members (Supplementary Table 3). However, we did not find any substantial differences in the 5' part of CSB II sequences, suggesting that the sequence of the CSB II G track does not influence the molecular outcomes in the analyzed patients. Consistent with the previously published data ⁸, we found that mutant TEFM-variants had a reduced ability to prevent premature termination at CSB II, particularly with templates containing longer (G)-tract sequences (Fig. 6C-E).

>Here the order is not very logical considering the part preceding this text block. It would be much more logical to start with the last sentence, 'Consistent with...etc' and then continue with 'Given that the composition.....etc' (komma removed).

We corrected the text on page 12:

“Consistent with the previously published data ⁸, we found that mutant TEFM-variants had a reduced ability to prevent premature termination at CSB II, particularly with templates containing longer (G)-tract sequences (Fig. 6C-E). Given that the composition of CSB II could have a confounding effect, we analysed the CSBII sequences in the patients and some of their family members (Supplementary Table 3). However, we did not find any substantial differences in the 5' part of CSB II sequences, suggesting that the sequence of the CSB II G track does not influence the molecular outcomes in the analysed patients.”

Pg17 553-555

The predicted length of the MTS in TEFM is 36 aa, it is possible that Arg34, which is the likely recognition site for the mitochondrial processing peptidase (MPP), is absolutely conserved in mammalian TEFM.

>This sentence doesn't flow and is non-sensical

We corrected this sentence and combined it with the next sentence:

“The predicted length of the MTS in TEFM is 36 aa. The mitochondrial processing peptidase (MPP) recognizes a large variety of MTSs in mitochondrial preproteins and cleaves a single site, often including arginine, at the -2 position^{37,38}. Therefore, it is possible that Arg34 is the likely recognition site for the MPP.”

Reviewer #4 (Remarks to the Author):

The authors provided a thorough response to the criticism and improved the manuscript.

However, an issue raised during the first round of reviews still needs to be addressed to strengthen the importance of the conclusions on TEFM detection in patients' tissues. Since there is no published information on anti-TEFM antibody specificity in Western Blot assays, the authors were requested to provide an experiment demonstrating the detection of recombinant TEFM and TEFM in any somatic cells (or mitochondrial extracts of these cells) with the antibodies used in experiments with patients' tissues. This figure could be presented just in the response to the reviewers' comments.

We used the same antibody that we used in our previous study ("TEFM (c17orf42) is necessary for transcription of human mtDNA", by Minczuk et al. PMID: 21278163). In this study the antibody was thoroughly validated and the figures below are copied from this publication.

Sub-cellular location of TEFM. The HOS cells were fractionated into fraction containing unbroken cells and cell debris ('D', lane 2), cytosol ('C', lane 3) and mitochondria ('M', lane 4) as described 'Materials and Methods' section. The protein fractions were analysed by western blotting using antibodies to endogenous TEFM. The location of TEFM was compared with that of the following marker proteins: TFAM (mitochondrial matrix), TOM22 (mitochondrial outer membrane), GAPDH (cytosol).

Defects in respiratory chain function upon TEFM gene silencing. (A) Western blot analyses of steady-state protein level of TEFM and subunits of respiratory chain complexes in control cells (untransfected cells and siRNA GFP) and cells treated with siRNA TEFM for 3 and 6 days.

REVIEWERS' COMMENTS

Reviewer #1 (Remarks to the Author):

I am satisfied with the revision and all my questions are answered

Reviewer #4 (Remarks to the Author):

The authors addressed all my concerns.